# Statistical Component Separation for Targeted Signal Recovery in Noisy Mixtures

**Bruno Régaldo-Saint Blancard**  *bregaldo@flatironinstitute.org*
*Center for Computational Mathematics*
*Flatiron Institute*
*New York, NY 10010*

**Michael Eickenberg**  *meickenberg@flatironinstitute.org*
*Center for Computational Mathematics*
*Flatiron Institute*
*New York, NY 10010*

**Reviewed on OpenReview:** *https://openreview.net/forum?id=OUWG6O4yo9*

## Abstract

Separating signals from an additive mixture may be an unnecessarily hard problem when one is only interested in specific properties of a given signal. In this work, we tackle simpler "statistical component separation" problems that focus on recovering a predefined set of statistical descriptors of a target signal from a noisy mixture. Assuming access to samples of the noise process, we investigate a method devised to match the statistics of the solution candidate corrupted by noise samples with those of the observed mixture. We first analyze the behavior of this method using simple examples with analytically tractable calculations. Then, we apply it in an image denoising context employing 1) wavelet-based descriptors, 2) ConvNet-based descriptors on astrophysics and ImageNet data. In the case of 1), we show that our method better recovers the descriptors of the target data than a standard denoising method in most situations. Additionally, despite not constructed for this purpose, it performs surprisingly well in terms of peak signal-to-noise ratio on full signal reconstruction. In comparison, representation 2) appears less suitable for image denoising. Finally, we extend this method by introducing a diffusive stepwise algorithm which gives a new perspective to the initial method and leads to promising results for image denoising under specific circumstances.

## 1 Introduction

We investigate the properties of a new class of source separation algorithms known as *statistical component separation methods* that has recently emerged for the analysis of astrophysical data (Regaldo-Saint Blancard et al., 2021; Delouis et al., 2022; Siahkoohi et al., 2023; Auclair et al., 2024). Contrary to standard source separation algorithms, such as blind source separation techniques (Cardoso, 1998), these methods do not focus on recovering the signal of interest, but on solely recovering certain statistics or features derived from this signal. These have proven successful in separating signals of distinct statistical natures in a variety of astrophysical contexts, such as the separation of interstellar dust emission and instrumental noise in data from the *Planck* satellite (Regaldo-Saint Blancard et al., 2021; Delouis et al., 2022), the separation of interstellar dust emission from the cosmic infrared background (Auclair et al., 2024), or the removal of glitches in seismic data from the *InSight* Mars mission (Siahkoohi et al., 2023). The methodology is not specific to astrophysics, and could be of interest to other scientific fields.

**Problem.** We observe a noisy mixture $y = x_0 + \epsilon_0$ with $x_0$ the signal of interest and $\epsilon_0$ a noise process. Signals can be viewed as vectors of $\mathbb{R}^M$. We refer to $\epsilon_0$ as the noise, but we make no assumption on its distribution $p(\epsilon_0)$, so that it can include any form of contaminant of the signal $x_0$. However, we assume that we have a way to sample $p(\epsilon_0)$. Let $\phi$ be a function, called the *representation*, that maps $x$ to a vector of features or summary statistics in $\mathbb{R}^K$, with typically $K \ll M$. The ultimate goal of a statistical component separation method is to recover $\phi(x_0)$. Regaldo-Saint Blancard et al. (2021) introduced a first algorithm to do that which consists in constructing $\hat{x}_0$ such that:

$$\hat{x}_0 \in \arg\min_x \mathcal{L}(x) \quad \text{with} \quad \mathcal{L}(x) = \mathbb{E}_{\epsilon \sim p(\epsilon_0)} \left[ \|\phi(x + \epsilon) - \phi(y)\|_2^2 \right], \tag{1}$$

and where the optimization of $\mathcal{L}$ is initialized with $y$. For suited $\phi$, previous works have demonstrated empirically that $\phi(\hat{x}_0)$ can be a relevant estimate of $\phi(x_0)$, and that, while not expected, $\hat{x}_0$ also seemed to be a reasonable estimate of $x_0$ (see Regaldo-Saint Blancard et al. (2021); Delouis et al. (2022); Siahkoohi et al. (2023); Auclair et al. (2024)).[1] The goal of this paper is to give first formal elements to explain these results, establish performance baselines, and introduce new methods for solving the problem. For numerical experiments, we will approximate the expected value involved in $\mathcal{L}$ by Monte Carlo estimates. Introducing $Q$ independent noise samples $\epsilon_1, \ldots, \epsilon_Q \sim p(\epsilon_0)$, the corresponding empirical loss $\hat{\mathcal{L}}$ reads:

$$\hat{\mathcal{L}}(x) = \frac{1}{Q} \sum_{i=1}^{Q} \|\phi(x + \epsilon_i) - \phi(y)\|^2. \tag{2}$$

**Related work.** Aside from the literature mentioned above, we are not aware of directly related work on this precise problem. However, we mention that adjacent problems of task-adapted reconstructions were explored in learning contexts. In particular, Mairal et al. (2012) investigated task-adapted dictionary learning, for which sparse data representations can be tuned to specific tasks. More recently, Adler et al. (2022) established a framework for task-related solving of inverse problems and showed how deep neural networks can be used for it. Finally, we add that the approach investigated in this paper shares similarities with sparse regularization techniques proposed in the extensive denoising and component separation literature (e.g., Starck et al., 2005; Selesnick, 2012; Elad et al., 2023). The loss $\mathcal{L}$ can indeed be interpreted as a regularization term in a denoising or component separation context. However, we emphasize that the goals differ, since statistical component separation methods primarily focus on recovering $\phi(x_0)$ (and not on recovering $x_0$).

**Outline.** In Sect. 2, we compute the analytical expressions of the global minimizers of $\mathcal{L}$ for different examples of representations $\phi$ and in the case of Gaussian noise $\epsilon_0$. This will give us a sense of the constraints that $\phi$ must respect for $\phi(\hat{x}_0)$ to be a relevant estimate of $\phi(x_0)$. In Sect. 3, we describe a first algorithm to perform numerical experiments in typical image denoising settings for two different representations: the first one based on wavelet phase harmonic statistical descriptors, and the second one based on ConvNet feature maps. Then, in Sect. 4, we discuss strategies to improve the results in the case of Gaussian noise using a new diffusive stepwise algorithm. We finally summarize our conclusions and perspectives in Sect. 5. Codes and data are provided on GitHub.[2]

**Notations.** We refer to the components of a vector $x \in \mathbb{C}^M$ as $x_i$ or $x[i]$. The dot product between $x$ and $y$ is $x \cdot y = \sum_{i=1}^{M} x_i \overline{y_i}$, and the corresponding norm of $x$ is $\|x\| = (\sum_{i=1}^{M} |x_i|^2)^{1/2}$. The convolution of $x, y \in \mathbb{C}^M$ is $x \star y$. We denote the matrix-vector product between $A$ and $x$ by $Ax$, or $A \cdot x$ when there is ambiguity. We call $\mathrm{sp}(A)$ the set of eigenvalues of a matrix $A$. For $A$ and $B$ two matrices of same size, the Frobenius inner product of $A$ and $B$ is $\langle A, B \rangle_{\mathrm{F}} = \mathrm{Tr}(A^\dagger B) = \sum_{i,j} \overline{A_{ij}} B_{ij}$, where $A^\dagger = \bar{A}^T$, and the corresponding norm of $A$ is $\|A\|_{\mathrm{F}} = \sum_{i,j} |A_{ij}|^2$. For $p_1, p_2$ two independent random processes, we write $p_1 \sim p_2$ when $p_1$ and $p_2$ follow the same distribution.

---

[1]Note that alternatives losses $\mathcal{L}$ have been considered in Delouis et al. (2022); Siahkoohi et al. (2023); Auclair et al. (2024), and have shown significant improvements for the estimation of $\phi(x_0)$, respectively. A formal investigation of these alternatives is left for future work.

[2]https://github.com/bregaldo/stat_comp_sep.

## 2 A First Analytical Exploration

As a first exploration, we compute the set of global minimizers of $\mathcal{L}$ as defined in Eq. (1) for simple examples of $\phi$, and choosing for $\epsilon_0$ a Gaussian white noise distribution of variance $\sigma^2$, that is $p(\epsilon_0) \sim \mathcal{N}(0, \sigma^2 I_M)$. This assumption for $\epsilon_0$ places us in a typical denoising framework. We also provide in App. B a discussion of how our method relates to maximum likelihood estimation in this context.

### 2.1 Linear Representation $\phi(x) = Ax$ leads to mean subtraction

To start with, we consider the case where $\phi(x) = Ax$, with $A$ an injective matrix of size $K \times M$ (with necessarily $K \geq M$). The vector $\phi(x)$ then simply consists in a set of features that are linear combinations of the input vector components. The following proposition establishes that the minimizer is necessarily $y - \mathbb{E}[\epsilon_0]$.

**Proposition 2.1.** *For $\phi(x) = Ax$ with $A$ injective, $\mathcal{L}$ has a unique global minimizer equal to $y - \mathbb{E}[\epsilon_0]$.*

This proposition is proven in App. A.1.1. It is a simple and informative result: if our representation is linear, then the minimization of $\mathcal{L}$ can only bring us back to the observation $y$ diminished by the mean of the noise. In particular, when the mean of the noise is zero, this optimization has no effect, prompting us to turn to a nonlinear operator $\phi$.

### 2.2 Quadratic Representation $\phi(x) = x^2$ leads to sqrt-thresholding

A very simple nonlinear representation is the pointwise quadratic function. Without loss of generality, we only consider the case where the dimension of $x$ is $M = 1$. The solution are given by the following proposition.

**Proposition 2.2.** *For $\phi(x) = x^2$ and $p(\epsilon_0) \sim \mathcal{N}(0, \sigma^2 I_M)$, the global minimizers of $\mathcal{L}$ are 0 when $y^2 \leq 3\sigma^2$, and $\pm\sqrt{y^2 - 3\sigma^2}$ when $y^2 > 3\sigma^2$.*

We refer the reader to App. A.1.2 for a proof. This solution introduces a threshold on $y^2$. If $y^2$ is not sufficiently large compared to the variance of the noise $\sigma^2$, then the solution is 0. Otherwise, we obtain $\pm\sqrt{y^2 - 3\sigma^2}$. This second case demonstrates an attempt to "subtract the noise magnitude from the signal magnitude".

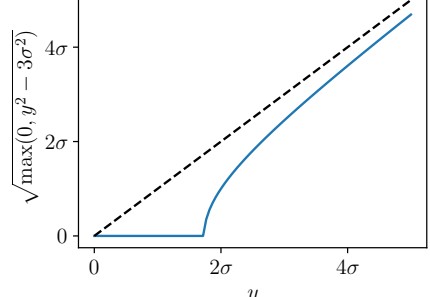

The behavior, shown in Fig. 1, is similar to that of the piecewise linear soft-thresholding operator. One significant difference is that it asymptotically reaches the identity function.

The intuition here is that, whenever the amplitude of the signal is too small compared to that of the noise, it is optimal to shrink it to zero. However, if the signal clearly stands out from the noise, then $y$ can be corrected accordingly through a small shrinkage.

Figure 1: sqrt-thresholding function involved in Sect. 2.2 (in blue), and asymptotic behavior (in black).

Note that related shrinkage functions are common in the context of sparse signal processing (see e.g. Selesnick, 2012; Al-Shabili et al., 2021).

### 2.3 Power Spectrum Representation

A standard summary statistic for stationary signals is the power spectrum - its distribution of power over frequency components. We consider signals of arbitrary dimension $M$, and a set of filters $\psi_1, \ldots, \psi_K$ typically well localized in Fourier space (i.e. "bandpass filters"). We assume that these filters cover Fourier space, meaning that for any Fourier mode $k$, there exists a $i_0$ such that $\hat{\psi}_{i_0}(k) \neq 0$.[3] We define the power spectrum representation as:

$$\phi(x) = \left(\|\psi_1 \star x\|^2, \ldots, \|\psi_K \star x\|^2\right). \tag{3}$$

---

[3]$\hat{\psi}$ here refers to the discrete Fourier transform of $\psi$.

It corresponds to a vector of $K$ coefficients measuring the power of the input signal on the passband of each of the filters $\psi_1, \ldots, \psi_K$.

If we assume that the frequency supports of these filters are disjoint, minimizing $\mathcal{L}$ is equivalent to minimizing independently for all $i = 1, \ldots, K$:

$$\mathcal{L}_i(x_i) = \mathbb{E}\left[\left(\|\Psi_i(x_i + \epsilon)\|^2 - \|\Psi_i y\|^2\right)^2\right], \tag{4}$$

where $x_i$ is the orthogonal projection of the input $x \in \mathbb{R}^M$ in the subspace spanned by the Fourier modes of the passband of $\psi_i$, and $\Psi_i$ is the injective matrix representing the linear operation $x \to \psi_i \star x$ in that subspace. The following proposition (proof given in App. A.1.3), determines the minimizers of $\mathcal{L}_i$.

**Proposition 2.3.** *For $\phi(x) = \|Ax\|^2$ with $A$ injective and $p(\epsilon_0) \sim \mathcal{N}(0, \sigma^2 I_M)$, introducing:*

$$\Lambda = \{\lambda \in \mathrm{sp}(A^T A) \text{ such that } \|Ay\|^2 - \mathbb{E}\left[\|A\epsilon\|^2\right] - 2\sigma^2\lambda \geq 0\}, \tag{5}$$

*if $\Lambda = \emptyset$, then the global minimizer of $\mathcal{L}$ is unique equal to 0, otherwise, the minimizers are the eigenvectors $x$ of $A^T A$ associated with $\min \Lambda$ such that $\|Ax\|^2 = \|Ay\|^2 - \mathbb{E}\left[\|A\epsilon\|^2\right] - 2\sigma^2 \min \Lambda$.*[4]

Let us take a step back before breaking down this result. The power spectrum is a statistic that is additive when computed on independent signals, so that for $a$ and $b$ two independent processes, we have $\mathbb{E}\left[\phi(a + b)\right] = \mathbb{E}\left[\phi(a)\right] + \mathbb{E}\left[\phi(b)\right]$. In our setting, where $x_0$ is viewed as a deterministic quantity, this gives:

$$\mathbb{E}_{\epsilon_0 \sim p(\epsilon_0)}\left[\phi(y)\right] = \phi(x_0) + \mathbb{E}_{\epsilon_0 \sim p(\epsilon_0)}\left[\phi(\epsilon_0)\right]. \tag{6}$$

Therefore, an unbiased estimator of the power spectrum statistics of $x_0$ based on the observation $y$ is:

$$\widehat{\phi(x_0)} = \phi(y) - \mathbb{E}_{\epsilon_0 \sim p(\epsilon_0)}\left[\phi(\epsilon_0)\right]. \tag{7}$$

Now, applied to $\phi_i(x_i) = \|\Psi_i x_i\|^2$, Prop. 2.3 tells us that there is a threshold below which the minimization of $\mathcal{L}_i$ leads to a signal that is zero over the passband of $\psi_i$, and above which this minimization leads to a signal $x_i$ such that $\|\Psi_i x_i\|^2 = \|\Psi_i y\|^2 - \mathbb{E}[\|\Psi_i \epsilon\|^2] - 2\sigma^2 \min \Lambda_i = \widehat{\phi_i(x_0)} - 2\sigma^2 \min \Lambda_i$. For typical filters, we usually have $2\sigma^2 \min \Lambda_i \ll \mathbb{E}[\|\Psi_i \epsilon\|^2]$, so that when the signal stands out from the noise, the global minimizers of $\mathcal{L}_i$ have power spectra statistics that almost coincide with the unbiased estimator of the power spectrum coefficients of $x_0$. In conclusion, in this setting, the power spectrum representation leads to minimizers $\hat{x}_0$ such that $\phi(\hat{x}_0)$ is an explicit estimate of $\phi(x_0)$.

We note that for filters with intersecting passbands, the previous analysis becomes significantly more technical. We prove in App. A.1.4 a generalization of Prop. 2.2 that addresses this case.

# 3 Statistical Component Separation and Image Denoising

Previous works have shown that statistical component separation methods can perform surprisingly well for image denoising provided that the representation $\phi$ is suited to the data (Regaldo-Saint Blancard et al., 2021; Delouis et al., 2022; Auclair et al., 2024). Although the goal of these methods remains to estimate $\phi(x_0)$, in this section, we investigate numerically to which extent $\hat{x}_0$ can also be a relevant estimate of $x_0$.

We focus on a typical denoising setting, where $\epsilon_0$ is a colored Gaussian stationary noise. We employ the block-matching and 3D filtering (BM3D) algorithm (Dabov et al., 2007; Mäkinen et al., 2020) as a benchmark. However, we emphasize that contrary to BM3D, statistical component separation methods can apply similarly to arbitrary noise processes, including non-Gaussian or non-stationary ones. This has already been illustrated in the previous literature on this subject, and we will consider an additional exotic noise in this section for this purpose.

We introduce in Sect. 3.1 the vanilla algorithm used for the experiments of this section. Then, we apply this algorithm for two distinct representations: Sect. 3.2 employs a representation based on the wavelet phase harmonics (WPH) statistics, and Sect. 3.3 uses summary statistics defined from feature maps of a ConvNet.

---

[4]We can verify that for $A$ a matrix of size $1 \times 1$, we recover a result equivalent to that of Prop. 2.2.

### 3.1 Vanilla Algorithm

---

**Algorithm 1** Vanilla Statistical Component Separation

---

**Inputs:** $y$, $p(\epsilon_0)$, $Q$, $T$, gradient-based optimizer (e.g. LBFGS)
**Initialize:** $\hat{x}_0 = y$
**for** $i = 1 \ldots T$ **do**
    **sample** $\epsilon_1, \ldots, \epsilon_Q \sim p(\epsilon_0)$
    $\hat{\mathcal{L}}(\hat{x}_0) = \sum_{k=1}^{Q} \left\| \phi(\hat{x}_0 + \epsilon_k) - \phi(y) \right\|^2 / Q$
    $\hat{x}_0 \leftarrow \text{ONE\_STEP\_OPTIM} \left[ \hat{x}_0, \nabla \hat{\mathcal{L}}(\hat{x}_0) \right]$
**end for**
**return** $\hat{x}_0$

---

Analytically determining the global minimizer of $\mathcal{L}$ for arbitrary representations $\phi$ quickly becomes intractable, in which case one has to solve this optimization problem numerically. A straightforward way to do that is to approximate $\mathcal{L}$ via Monte Carlo estimates. Introducing $Q$ independent noise samples $\epsilon_1, \ldots, \epsilon_Q \sim p(\epsilon_0)$, we define:

$$\hat{\mathcal{L}}(x) = \frac{1}{Q} \sum_{i=1}^{Q} \left\| \phi(x + \epsilon_i) - \phi(y) \right\|^2. \tag{8}$$

The minimization of $\mathcal{L}$ then takes the form of a regular stochastic optimization described in Algorithm 1, and referred to as the *vanilla* algorithm in the following. This algorithm was used in Regaldo-Saint Blancard et al. (2021), where the authors had employed a L-BFGS optimizer (Byrd et al., 1995; Zhu et al., 1997) using $y$ as the initial guess. We proceed similarly in the following, and fix the number of iterations to $T = 30$ and the batch size to $Q = 100$.[5]

### 3.2 Wavelet Phase Harmonics Representation

**Definition.** Similarly to Regaldo-Saint Blancard et al. (2021); Auclair et al. (2024), we consider a representation based on wavelet phase harmonics (WPH) statistics (Mallat et al., 2019; Zhang & Mallat, 2021; Allys et al., 2020). These statistics efficiently capture coherent structures in a variety of non-Gaussian stationary data. They rely on the wavelet transform, which locally decomposes the signal onto oriented scales. Formally, for a random image $x$, the WPH statistics are estimates of covariances between pointwise nonlinear transformations of the wavelet transform of $x$. Using a set of complex-valued wavelets $\psi_1, \ldots, \psi_N$ covering Fourier space with their respective passbands, we focus on covariances of the form:

$$S_i^{11}(x) = \text{Cov}\left[x \star \psi_i, x \star \psi_i\right], \qquad\qquad S_i^{00}(x) = \text{Cov}\left[|x \star \psi_i|, |x \star \psi_i|\right], \tag{9}$$

$$S_i^{01}(x) = \text{Cov}\left[|x \star \psi_i|, x \star \psi_i\right], \qquad\qquad C_{i,j}^{01}(x) = \text{Cov}\left[|x \star \psi_i|, x \star \psi_j\right]. \tag{10}$$

We give in App. D the technical details related to the definition and computation of these statistics. In practice, $\phi(x)$ is made of $K = 420$ complex-valued coefficients for $x$ a $M = 256 \times 256$ image.

**Experimental Setting.** We consider three different types of $256 \times 256$ images corresponding to a simulation of the emission of dust grains in the interstellar medium (the *dust* image), a simulation of the large-scale structure of the Universe (Villaescusa-Navarro et al., 2020) (the *LSS* image), and randomly selected images from the ImageNet dataset (Deng et al., 2009) (the *ImageNet* images). We give additional details on this data in App. C. We then consider four different noise processes: three colored Gaussian noises, namely pink, white, and blue noises, as well as a non-Gaussian noise made of small crosses (see Fig. 2, bottom right). We vary, for the colored noises, the amplitude $\sigma$ of the noise considering 10 different levels ranging from 0.1 to 2.14 (logarithmically spaced) in unit of the standard deviation of $x$, and for the "crosses" noises, the density of crosses $\rho$ considering 10 different values ranging from 0.001 to 0.063 (logarithmically spaced)[6]. For each of

---

[5]We have found empirically that the batch size should be kept sufficiently high for the L-BFGS optimizer to behave correctly. We also report that $T = 30$ was sufficient to achieve approximate convergence for all experiments.

[6]The density $\rho$ is defined as the expected ratio of crosses to the total number of pixels.

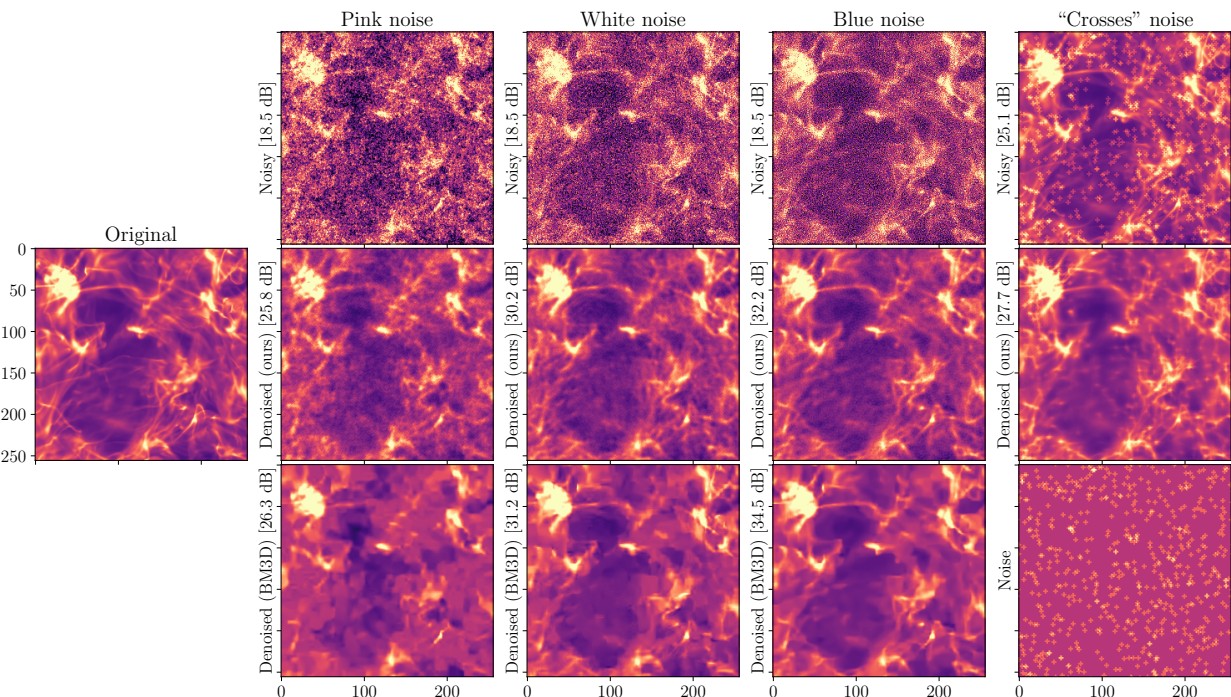

Figure 2: Original dust image $x_0$ (center left), noisy realizations $y$ for distinct colored Gaussian noise processes and a non-Gaussian noise (top row), denoised images $\hat{x}_0$ using Algorithm 1 with the WPH representation from Sect. 3.2 (middle row), and denoised images using BM3D for colored noises (bottom row, except bottom right). Additionally, a sample of the non-Gaussian "crosses" noise is shown (bottom right). The accompanying PSNR values are provided next to each noisy and denoised image.

theses cases, we apply Algorithm 1 for 30 different noise realizations. Each optimization takes $\sim 40$ s with a GPU-accelerated code on a A100 GPU.

**Results.** In Fig. 2, we compare the noise-free dust image $x_0$ with examples of its noisy $y$ and denoised $\hat{x}_0$ versions for each noise process. In the case of the colored noises, we also include the BM3D-denoised images in the bottom row for reference. Our method effectively reduces noise while preserving the original image's structure. Even with the "crosses" noise, our algorithm successfully removes most of the crosses, the remaining ones having been most likely confused with actual structures of $x_0$. We evaluate the quality of the denoising in terms of peak signal-to-noise ratio (PSNR), significantly improving it in all cases. However, BM3D outperforms our method for colored noises, which is not surprising as our approach was not explicitly designed for that.

We further evaluate our algorithm's performance for colored noises using PSNR and the relative error of $\phi(\hat{x}_0)$. In Fig. 3, we present the PSNR and relative errors of WPH statistics for different coefficient classes as a function of noise level $\sigma$. Our method consistently improves PSNR, but BM3D performs better across all noise levels. We note that our method's performance degrades for very high $\sigma$, potentially due to structure hallucination caused by extreme initial noise.[7] In terms of WPH statistics relative error, our method effectively reduces noise impact and outperforms BM3D in most cases. Notably, it excels in $S^{11}$ and $S^{01}$ coefficients, except for the high-noise regime in $S^{11}$. However, for $S^{00}$ and $C^{01}$ coefficients, in the cases of blue and white noise, BM3D performs comparably or better than our method. Since a perfect PSNR implies perfect statistics recovery, we interpret this as the sign that a regular denoising algorithm, when adapted to the noise process, can also provide precise estimates of $\phi(x_0)$. However, we point out that the normalization of the WPH statistics may play a crucial role on these metrics (see App. D), and a fairer comparison should explore

---

[7]WPH statistics may also define generative models with a sampling procedure sharing important similarities with Algorithm 1 (see Allys et al. (2020); Zhang & Mallat (2021); Regaldo-Saint Blancard et al. (2021); Régaldo-Saint Blancard et al. (2023)).

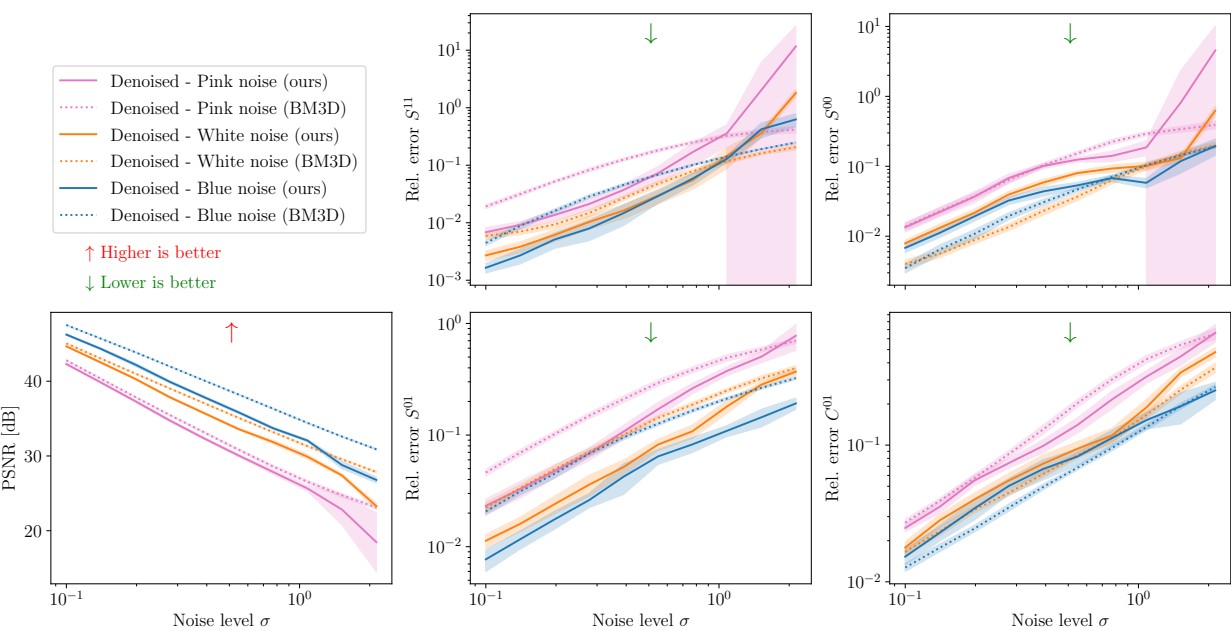

Figure 3: PSNR and relative errors of the WPH statistics per class of coefficients as a function of the noise level $\sigma$ for the denoised dust images as described in Sect. 3.2 for each type of colored noise.

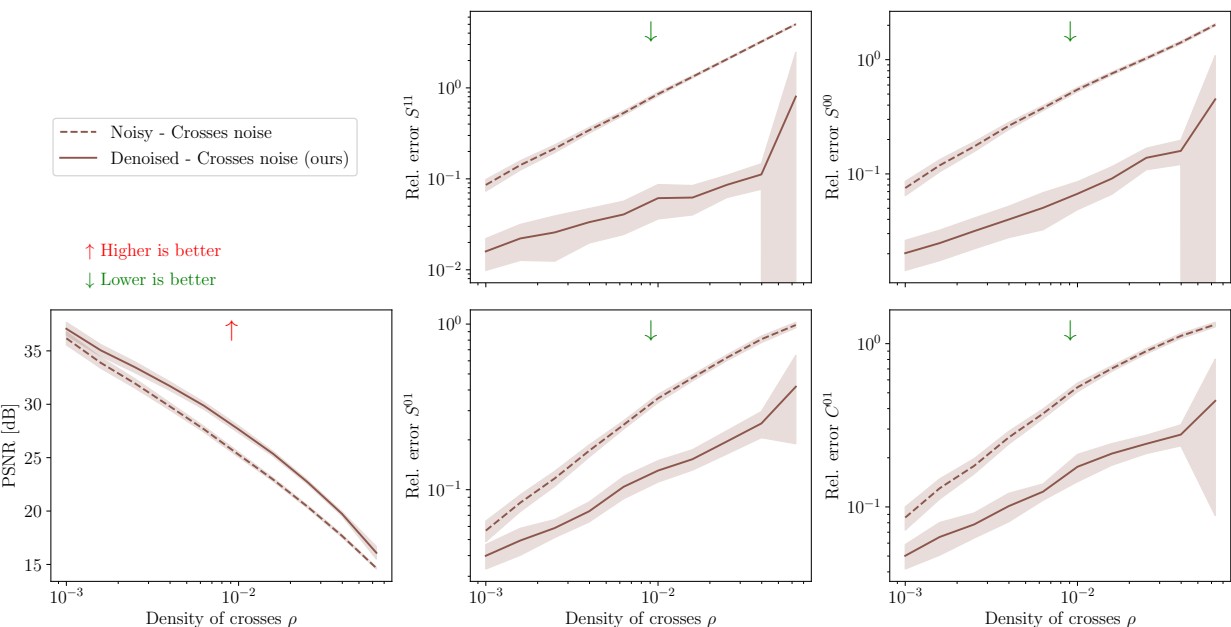

Figure 4: Same as Fig. 2 for the "crosses" noises as a function of the density of crosses $\rho$.

its role in this precise setting. We report similar results in Fig. 4 for the "crosses" noise. These demonstrate a clear mitigation of the noise in all regimes.

We report in Figs. F.1, F.2, F.4, F.5, and F.7 equivalent results for the LSS and ImageNet data. The overall conclusions are the same.

### 3.3 ConvNet-based Representation

**Definition.** To further explore the dependence on the representation $\phi$ for the performance of a statistical component separation for image denoising, we define a representation based on feature maps of a ConvNet. We make use of the VGG-19_BN network (Simonyan & Zisserman, 2015) which was trained for image classification on ImageNet. We only employ the first two convolutional blocks of the networks (each of these being made of a sequence of `Conv2d`, `BatchNorm2d`, and `ReLU` blocks), which, for each $224 \times 224$ RGB image $x$, output a set of 64 $224 \times 224$ feature maps $f(x) = (f_1(x), \ldots, f_{64}(x))$. We define a representation $\phi(x)$ of dimension $K = 64$ from these feature maps by taking the squared Euclidean norm of each feature map, that is:

$$\phi(x) = (\|f_1(x)\|^2, \ldots, \|f_{64}(x)\|^2). \tag{11}$$

This representation shares structural similarities with the WPH statistics and its parent the wavelet scattering transform statistics (Bruna & Mallat, 2013; Mallat, 2016). However, contrary to these previous transforms which employ generic wavelet filters, the filters of the VGG convolutional layers are specifically trained for the analysis of ImageNet data.

**Results.** We conduct the same experiment as in Sect. 3.2 in this setting focusing on randomly selected ImageNet images. We show in Fig. F.8 examples of resulting images before and after denoising, and we show in Fig. 5 the root-mean-square error on the coefficients of the representation $\phi$ as a function of $\sigma$ (mean and standard deviation computed across 50 randomly selected test images). While Fig. 5 indicates that our method significantly reduces the impact of the noise on the representation $\phi$, we see in Fig. F.8 that it performs poorly as a regular denoiser in comparison to BM3D. This is an interesting result, as it shows that the representation $\phi$ we have built here is not as well suited for image denoising as the WPH representation was. Given the fact that VGG networks were trained for a classification problem, it is likely that their feature maps are partly robust to the noise in the input. A spectral analysis of the denoised maps further shows that there remains a significant amount of noise in the small scales, suggesting that the feature maps are weakly impacted by the noise at small scales.

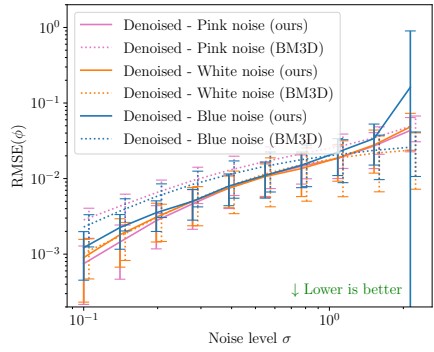

Figure 5: Root-mean-square error on the coefficients of the ConvNet-based representation $\phi$ as a function of the noise level $\sigma$ for the denoised ImageNet images for each type of colored noise.

## 4 Diffusive Statistical Component Separation

Taking inspiration from the diffusion-based generative modeling literature (Ho et al., 2020; Song et al., 2021), we investigate to which extent statistical component separation methods can benefit from the idea of breaking down the optimization into a sequence of optimization problems involving noises of smaller amplitude.

We introduce in Sect. 4.1 a new algorithm that leverages this idea in the case of Gaussian noises and apply it to the dust data introduced in Sect. 3. We then study in Sect. 4.2 the limit regime where the amplitude of the noise tends to zero. This gives us an alternative way to perform statistical component separation giving promising results in specific circumstances.

### 4.1 A "Diffusive" Algorithm

**Stable process.**   We assume that $\epsilon_0$ is a stable noise process, that is for any $\epsilon_1, \ldots, \epsilon_P \overset{i.i.d.}{\sim} p(\epsilon_0)$, we have $\sum_{i=1}^{P} \epsilon_i \sim \alpha \epsilon_0 + \beta$ for some scalar constants $\alpha$ and $\beta$. A practical example is that of Gaussian processes, since for $\epsilon_0 \sim \mathcal{N}(0, \Sigma)$, we clearly have $\sum_{i=1}^{P} \epsilon_i \sim P\epsilon_0 \sim \mathcal{N}(0, P\Sigma)$. Introducing $\alpha \in \mathbb{R}^P$ such that $\alpha_i > 0$ for all $i$, and $\|\alpha\|^2 = \sum_i \alpha_i^2 = 1$, and provided that $\mathbb{E}\left[\epsilon_0\right] = 0$, we can break down $\epsilon_0$ into "smaller" independent noise processes as follows:

$$\epsilon_0 = \sum_{i=1}^{P} \alpha_i \epsilon_i, \text{ with } \epsilon_1, \ldots, \epsilon_P \overset{i.i.d.}{\sim} p(\epsilon_0), \tag{12}$$

where the variance of $\alpha_i \epsilon_i$ can be made arbitrarily small by taking a sufficiently small value for $\alpha_i$.

**Algorithm.**   In this setting, we leverage this decomposition by breaking down the minimization of $\mathcal{L}$ into "simpler" optimization problems involving noises of smaller variance, with the goal of finding a better optimum $\hat{x}_0$. We introduce Algorithm 2 for this purpose. This algorithm starts from $\hat{x}_P = y$ and builds a sequence of signals $\hat{x}_{P-1}, \ldots, \hat{x}_0$ such that for all $i \in \{P-1, \ldots, 0\}$:

$$\hat{x}_i \in \arg\min_x \mathcal{L}(x; \alpha_{i+1}, \hat{x}_{i+1}) = \mathbb{E}_{\epsilon \sim p(\epsilon_0)}\left[\|\phi(x + \alpha_{i+1}\epsilon) - \phi(\hat{x}_{i+1})\|^2\right]. \tag{13}$$

---

**Algorithm 2** Diffusive Statistical Component Separation

> **Input:** $y$, $p(\epsilon_0)$, $Q$, $T$, $P$, $\alpha \in \mathbb{R}^P$ with $\|\alpha\|^2 = 1$ and $\alpha_i > 0$, gradient-based optimizer
> **Initialize:** $\hat{x}_P = y$
> **for** $i = P - 1 \ldots 0$ **do**
>     $\hat{x}_i = \hat{x}_{i+1}$
>     **for** $j = 1 \ldots T$ **do**
>         **sample** $\epsilon_1, \ldots, \epsilon_Q \sim p(\epsilon_0)$
>         $\hat{\mathcal{L}}(\hat{x}_i) = \sum_{k=1}^{Q} \|\phi(\hat{x}_i + \alpha_{i+1}\epsilon_k) - \phi(\hat{x}_{i+1})\|^2 / Q$
>         $\hat{x}_i \leftarrow \text{ONE\_STEP\_OPTIM}\left[\hat{x}_i, \nabla\hat{\mathcal{L}}(\hat{x}_i)\right]$
>     **end for**
> **end for**
> **return** $\hat{x}_0$

---

A sufficient condition for a perfect reconstruction to be achieved is that $\hat{x}_i \approx x_0 + \tilde{\epsilon}_i$ with $\tilde{\epsilon}_i \sim (1 - \sum_{j=0}^{P-i-1} \alpha_{P-j}^2)^{1/2} \epsilon_0$. We do not expect this strong condition to hold for arbitrary functions $\phi$, but the design of $\phi$ should be guided in that sense.

We also note that a stepwise approach had also been employed in Delouis et al. (2022), and we draw some connections between the two approaches in App. E. We show that these two approaches rely on related objective functions. A further exploration of the pros and cons of each of these two algorithms is left for future work.

**Experiment.**   We apply Algorithm 2 to the dust data introduced in Sect. 3 in the case of the Gaussian white noise and the WPH representation with $\alpha_i = 1/\sqrt{P}$ and $P = \lfloor 10\sigma \rfloor$. We compare in Fig. 6 the results of this algorithm to those of Algorithm 1. We see that this approach slightly improves the results for every metric except for the relative error on the $C^{01}$ coefficients at an intermediate noise level where these are slightly deteriorated. Although these numerical experiments are not showing significant improvements, having at least consistent results suggest that, from a theoretical perspective, component separation methods can be understood as the aggregation of these small optimization problems. We push this idea further in the rest of this section.

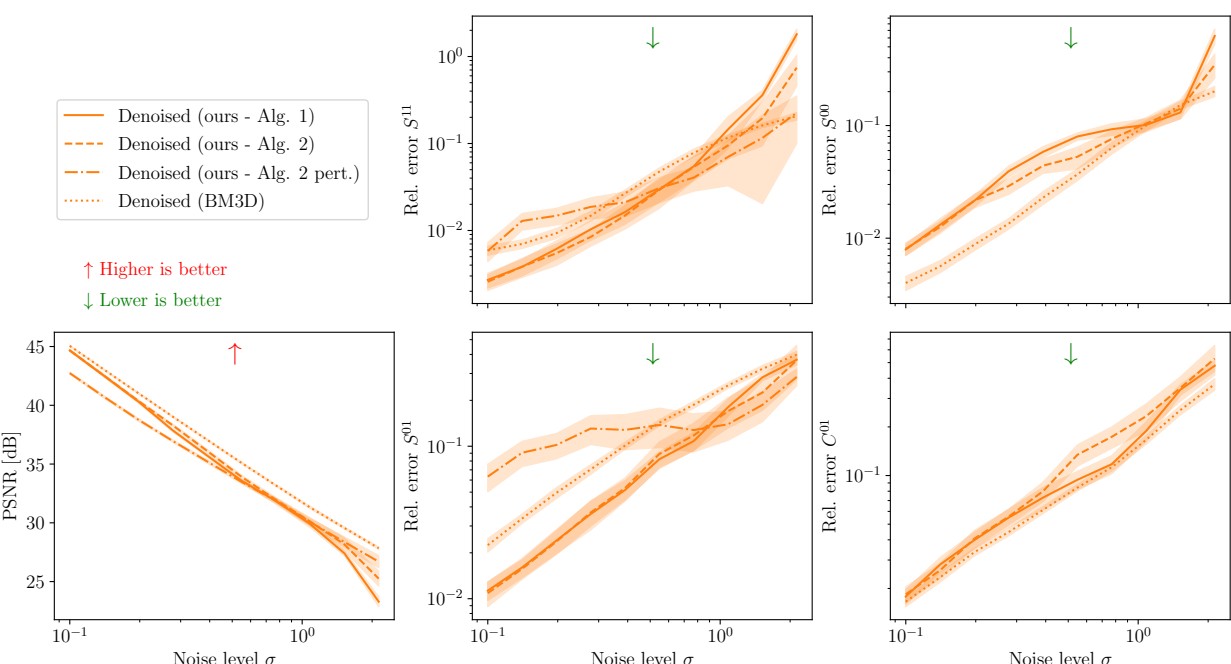

Figure 6: Same as Fig. 3 for the white noise case only, and comparing results obtained with Algorithm 1 (see Sect. 3.2), Algorithm 2 with a non-perturbative loss (see Sect. 4.1), Algorithm 2 with the perturbative loss introduced in Sect. 4.3, and BM3D.

## 4.2 Limit Regime for Infinitely Small Noises

We study the limit regime of infinitely small noises to give a more formal perspective to Algorithm 2. We introduce $\mathcal{L}(x, \alpha) = \mathbb{E}\left[\|\phi(x + \alpha\epsilon) - \phi(y)\|^2\right]$ and expands it with respect to $\alpha \in \mathbb{R}$.

**Proposition 4.1.** *For $\phi$ a twice differentiable function, and $\epsilon_0$ arbitrarily distributed with $\mathbb{E}[\epsilon_0] = 0$, we have:*

$$\mathcal{L}(x, \alpha) = \|\phi(x) - \phi(y)\|^2 + \alpha^2 \left[\langle J_\phi(x)^T J_\phi(x), \Sigma\rangle_{\mathrm{F}} + \langle H_\phi(x), \Sigma\rangle_{\mathrm{F}} \cdot (\phi(x) - \phi(y))\right] + o(\alpha^2), \tag{14}$$

*where $J_\phi(x)$ is the Jacobian matrix of $\phi$ (of size $K \times M$), $H_\phi(x)$ is its Hessian tensor (of rank 3, and size $K \times M \times M$), and $\Sigma$ is the covariance matrix of $\epsilon_0$.[8]*

The proof is given in App. A.2.1. The zeroth-order term of this expansion has a clear interpretation. It prevents the representation of $x$ from moving too far from that of $y$. However, the second order term is more intricate as it combines first and second order derivatives of $\phi$, which are weighted by the covariance of the noise. As an example, let us consider $\Sigma = \sigma^2 I_M$. We then get:

$$\langle J_\phi(x)^T J_\phi(x), \Sigma\rangle_{\mathrm{F}} = \sigma^2 \|J_\phi(x)\|_{\mathrm{F}}^2, \tag{15}$$

$$\langle H_\phi(x), \Sigma\rangle_{\mathrm{F}} \cdot (\phi(x) - \phi(y)) = \sigma^2 \mathrm{Tr}\left[H_\phi(x)\right] \cdot (\phi(x) - \phi(y)). \tag{16}$$

The first term is proportional to the squared norm of the Jacobian matrix, while the second term is a dot product between the vector of Hessian traces with the vector $\phi(x) - \phi(y)$. The trace of $H_{\phi_i}(x)$ directly relates to the mean curvature of the function $\phi_i$ at point $x$, so that this dot product quantifies the alignment between $\phi(x) - \phi(y)$ and the vector of mean curvatures for each component of $\phi$.

---

[8]We denote by $\langle H_\phi(x), \Sigma\rangle_{\mathrm{F}}$ the vector of $\mathbb{R}^K$ such that component $i$ equals $\langle H_{\phi_i}(x), \Sigma\rangle_{\mathrm{F}}$.

### 4.3 A Perturbative Algorithm

We note that contrary to Eq. (1), Eq. (14) got rid of the expected value over $\epsilon$, and the second-order expansion of $\mathcal{L}(x, \alpha)$ now only depends on the covariance matrix $\Sigma$ of the noise, which is often known or easy to estimate from a set of noise samples. Then, provided $\mathcal{L}(x, \alpha)$ can be correctly approximated by its second-order expansion in $\alpha$, one can evaluate the loss in a much more straightforward way that does not rely on a noisy Monte Carlo estimate. In this limit regime, the computational challenge lies in efficiently computing the second-order term of Eq. (14) in a differentiable way.

We compute in App. A.3 the relevant analytical expressions to do so for $\phi$ the WPH representation introduced in Sect. 3.2. We then apply Algorithm 2 for the dust image in the Gaussian white noise case with $\alpha_i = 1/\sqrt{P}$, $P = \lfloor 10\sigma \rfloor$, and $T = 10$, and by replacing the loss $\mathcal{L}$ with by its truncated second-order Taylor expansion as explicited in Eq. (14). We empirically found that the most stable setting is the one where the WPH statistics only include the $S^{11}$ and $S^{01}$ coefficients. We report in Fig. 6 the quantitative results in this setting for the PSNR and relative errors on these coefficients as a function of $\sigma$. Our method performs remarkably well in comparison to the previous results. It is very close to BM3D in terms of PSNR and outperforms it in the high noise regime in terms of relative errors on the $S^{11}$ and $S^{01}$ coefficients. The fact that the relative error for the $S^{01}$ coefficients is higher than for the noisy data in the low noise regime however suggests a form of instability for this range of $\sigma$. Nevertheless, we find these results promising and a clear demonstration of the relevance of this "diffusive" perspective on statistical component separation methods.

## 5 Conclusion

This paper has explored several aspects of statistical component separation methods. Section 2 has exhibited analytically the global minimizers of $\mathcal{L}$ for several examples of representations $\phi$ in the case of Gaussian white noise. We have shown that a linear $\phi$ cannot extract any information on $x_0$, while a simple quadratic representation leads to a form of `sqrt`-thresholding of the observation $y$. For $\phi$ a power spectrum representation, which can be viewed as a more general quadratic representation, the minimizers of $\mathcal{L}$ lead to relevant estimations of $\phi(x_0)$. Then, in Sect. 3, we have approached numerically the minimizers of $\mathcal{L}$ introducing Algorithm 1 in an image denoising setting for two representations where analytical calculations are intractable: 1) WPH statistics, 2) statistics derived from the feature maps of a ConvNet. For 1), Algorithm 1 acts as a regular image denoiser while not explicitly constructed for this. Although it does not outperform BM3D in terms of the PSNR metric, it better recovers the coefficients $\phi(x_0)$ for most classes of coefficients and experimental settings. Additionally, it may extend to arbitrary noise processes as it was illustrated with an exotic noise made of small crosses. For 2), the impact on the noise was clearly mitigated in $\phi(\hat{x}_0)$, but the resulting images $\hat{x}_0$ were still very noisy, showing that this representation is less suited for image denoising. Finally, in Sect. 4, we have introduced Algorithm 2, a "diffusive" statistical component separation method that can be applied in contexts where the noise is a stable process. These ideas led in some cases to better results than Algorithm 1 in a denoising setting. And more importantly, it supported the idea that statistical component separation methods can be described as the sequence of optimization problems with noises of smaller amplitudes. This idea will be pushed further in future work.

### Acknowledgments

It is a pleasure to thank Erwan Allys, Jean-Marc Delouis, Stéphane Mallat, Loucas Pillaud-Vivien, and Léo Vacher for valuable discussions.

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

# A  Proofs

## A.1  Proofs of Sect. 2

We give the proofs of the results presented in Sect. 2. Assuming that $\epsilon_0$ is a Gaussian white noise of variance $\sigma^2$, that is $p(\epsilon_0) \sim \mathcal{N}(0, \sigma^2 I_M)$, we compute the set of global minimizers of $\mathcal{L}$ as defined in Eq. (1) in the following cases:

1. $\phi(x) = Ax$ with $A$ an injective matrix of size $K \times M$,

2. $\phi(x) = x^2$ for $M = 1$,

3. $\phi(x) = \|Ax\|^2$ with $A$ an injective matrix of size $K' \times M$,

4. $\phi(x) = (\|A_1\|^2, \ldots, \|A_K\|^2)$ with $A_1, \ldots, A_K$ $K$ injective matrices of size $K' \times M$ such that $A_1^T A_1, \ldots, A_K^T A_K$ are co-diagonalizable.

For each case, note that $\mathcal{L}(x)$ is infinitely differentiable and obviously bounded from below by 0, and we will show below that $\lim\limits_{x \to +\infty} \mathcal{L}(x) = +\infty$ to demonstrate the existence of a global minimum.

We will then determine the global minimizers by studying the zeros of the gradient of $\mathcal{L}$, which, in its general form, reads:

$$\nabla \mathcal{L}(x) = 2 \mathbb{E}_{\epsilon \sim p(\epsilon_0)} \left[ J_\phi(x + \epsilon)^T \cdot (\phi(x + \epsilon) - \phi(y)) \right], \tag{17}$$

where $J_\phi$ is the Jacobian matrix of $\phi$.

### A.1.1  $\phi(x) = Ax$ with $A$ injective

**Proposition 2.1.** *For $\phi(x) = Ax$ with $A$ injective, $\mathcal{L}$ has a unique global minimizer equal to $y - \mathbb{E}[\epsilon_0]$.*

*Proof.* We have:

$$\mathcal{L}(x) = \mathbb{E} \left[ \|A(x + \epsilon - y)\|^2 \right]. \tag{18}$$

If $A$ is injective of size $K \times M$ (in which case, we necessarily have $K \geq M$), $A^T A$ is positive-definite and we call $\lambda_{\min} > 0$ its smallest eigenvalue. We have $\|Ax\|^2 \geq \lambda_{\min} \|x\|^2$, so that:

$$\mathcal{L}(x) \geq \lambda_{\min} \mathbb{E} \left[ \|x + \epsilon - y\|^2 \right] = \lambda_{\min} \|x\|^2 + O(\|x\|). \tag{19}$$

This shows that $\lim\limits_{x \to +\infty} \mathcal{L}(x) = +\infty$ so that a global minimum does exist.

Now, noticing that $J_\phi(x) = A$, we have:

$$\nabla \mathcal{L}(x) = 2A^T \cdot \mathbb{E}\left[\phi(x + \epsilon) - \phi(y)\right], \tag{20}$$

$$= 2A^T \cdot \phi(\mathbb{E}\left[x + \epsilon - y\right]), \tag{21}$$

$$= 2A^T \cdot \phi(x - y + \mathbb{E}\left[\epsilon\right]), \tag{22}$$

$$= 2A^T A \cdot (x - y + \mathbb{E}\left[\epsilon\right]). \tag{23}$$

For injective $A$, $A^T A$ is invertible, and we have $\nabla \mathcal{L}(x) = 0 \iff x = y - \mathbb{E}[\epsilon]$. This shows that, in this case, the only global minimizer is $y - \mathbb{E}[\epsilon]$.

Note that this result is independent of the noise distribution $p(\epsilon_0)$. $\qquad\square$

This result can be extended to non-injective matrices $A$. In this case, there is an affine subspace of solutions with the same loss value along shifts within the null space of $A$. By a similar procedure as above, we obtain that $x = y - \mathbb{E}[\epsilon] + v$, where $v \in \ker A$. Non-injectivity does not change the overall conclusion that, besides the subtraction of the mean of the noise, denoising does not take place.

**A.1.2** $\phi(x) = x^2$ **for** $M = 1$

**Proposition 2.2.** *For $\phi(x) = x^2$ and $p(\epsilon_0) \sim \mathcal{N}(0, \sigma^2 I_M)$, the global minimizers of $\mathcal{L}$ are 0 when $y^2 \leq 3\sigma^2$, and $\pm\sqrt{y^2 - 3\sigma^2}$ when $y^2 > 3\sigma^2$.*

*Proof.* For $\phi(x) = x^2$ and $K = M = 1$, we can show by expanding $\mathcal{L}$ that:

$$\mathcal{L}(x) = \|x\|^4 + O(\|x\|^2) \xrightarrow[x \to +\infty]{} +\infty. \tag{24}$$

This proves the existence of a global minimum.

Now, injecting $\phi'(x) = 2x$ in Eq. (17), we get:

$$\mathcal{L}'(x) = 4\mathbb{E}\left[(x + \epsilon)((x + \epsilon)^2 - y^2)\right], \tag{25}$$

$$= 4\mathbb{E}\left[(x + \epsilon)^3 - (x + \epsilon)y^2\right], \tag{26}$$

$$= 4\mathbb{E}\left[x^3 + \epsilon^3 + 3x^2\epsilon + 3x\epsilon^2 - xy^2 - \epsilon y^2\right], \tag{27}$$

$$= 4\left(x^3 + x(3\sigma^2 - y^2)\right), \tag{28}$$

where we have used the fact that $\mathbb{E}[\epsilon] = \mathbb{E}[\epsilon^3] = 0$ and $\mathbb{E}[\epsilon^2] = \sigma^2$. The zeros of $\mathcal{L}'$ are thus $x = 0$ or $x = \pm\sqrt{y^2 - 3\sigma^2}$ when $y^2 \geq 3\sigma^2$.

By looking at the values of $\mathcal{L}''(x) = 12x^2 + 12\sigma^2 - 4y^2$ at these points, we find that whenever $y^2 > 3\sigma^2$, $\mathcal{L}''(0) < 0$ so that 0 cannot be a local minimizer. In this case, we only have $\pm\sqrt{y^2 - 3\sigma^2}$ left to be the global minimizers.

All of this shows that the global minimizers of $\mathcal{L}$ are 0 when $y^2 \leq 3\sigma^2$, and $\pm\sqrt{y^2 - 3\sigma^2}$ when $y^2 > 3\sigma^2$. $\square$

This denoising function $x = \text{sgn}(y)\sqrt{\max(0, y^2 - 3\sigma^2)}$ is similar to the soft-thresholding function $\text{st}_\lambda(y) = \text{sgn}(y)\max(|y - \lambda|, 0)$, sharing the flat region around the origin. The main difference is that it contrary to soft-thresholding, this function approaches the identity function for high values. It also has infinite derivatives at the border of its flat region.

**A.1.3** $\phi(x) = \|Ax\|^2$ **with** $A$ **injective**

**Proposition 2.3.** *For $\phi(x) = \|Ax\|^2$ with $A$ injective and $p(\epsilon_0) \sim \mathcal{N}(0, \sigma^2 I_M)$, introducing:*

$$\Lambda = \{\lambda \in \text{sp}(A^T A) \text{ such that } \|Ay\|^2 - \mathbb{E}\left[\|A\epsilon\|^2\right] - 2\sigma^2\lambda \geq 0\}, \tag{29}$$

*if $\Lambda = \emptyset$, then the global minimizer of $\mathcal{L}$ is unique equal to 0, otherwise, the minimizers are the eigenvectors $x$ of $A^T A$ associated with $\min \Lambda$ such that $\|Ax\|^2 = \|Ay\|^2 - \mathbb{E}\left[\|A\epsilon\|^2\right] - 2\sigma^2 \min \Lambda$.*[9]

*Proof.* With $A$ an injective matrix of size $K' \times M$, we have:

$$\mathcal{L}(x) = \mathbb{E}\left[\left(\|A(x + \epsilon)\|^2 - \|Ay\|^2\right)^2\right], \tag{30}$$

$$= \mathbb{E}\left[\|A(x + \epsilon)\|^4 + \|Ay\|^4 - 2\|A(x + \epsilon)\|^2\|Ay\|^2\right]. \tag{31}$$

Writing that $\|A(x + \epsilon)\|^2 = \|Ax\|^2 + \|A\epsilon\|^2 + 2Ax \cdot A\epsilon$, we can show that:

$$\mathbb{E}\left[\|A(x + \epsilon)\|^2\right] = \|Ax\|^2 + \mathbb{E}\left[\|A\epsilon\|^2\right], \tag{32}$$

$$\mathbb{E}\left[\|A(x + \epsilon)\|^4\right] = \|Ax\|^4 + \mathbb{E}\left[\|A\epsilon\|^4\right] + 4\sigma^2\|A^T Ax\|^2 + 2\|Ax\|^2\mathbb{E}\left[\|A\epsilon\|^2\right], \tag{33}$$

---

[9]We can verify that for $A$ a matrix of size $1 \times 1$, we recover a result equivalent to that of Prop. 2.2.

where we have used the facts that $\mathbb{E}[A\epsilon] = 0$ and $\mathbb{E}[\epsilon\epsilon^T] = \sigma^2 I_M$. We can then rewrite $\mathcal{L}$ as follows:

$$\mathcal{L}(x) = \|Ax\|^4 + 4\sigma^2 \|A^T Ax\|^2 + 2\|Ax\|^2 \left(\mathbb{E}\left[\|A\epsilon\|^2\right] - \|Ay\|^2\right) + \mathcal{L}(0). \tag{34}$$

Same as Sect. A.1.1, with $\lambda_{\min} = \min \mathrm{sp}(A^T A) > 0$, where $\mathrm{sp}(A^T A)$ is the spectrum of $A^T A$, we can show that:

$$\mathcal{L}(x) \geq \lambda_{\min}^2 \|x\|^4 + O(\|x\|^2), \tag{35}$$

which leads to $\lim\limits_{x \to +\infty} \mathcal{L}(x) = +\infty$ and proves the existence of a global minimum.

Starting from Eq. (34) and using the fact that $\nabla\phi(x) = 2A^T Ax$, we show that:

$$\nabla\mathcal{L}(x) = 4\left(\|Ax\|^2 + \mathbb{E}\left[\|A\epsilon\|^2\right] - \|Ay\|^2\right)A^T Ax + 8\sigma^2(A^T A)^2 x. \tag{36}$$

Since $A^T A$ is an invertible matrix, we then have:

$$\nabla\mathcal{L}(x) = 0 \iff (\alpha(x)I_M + A^T A)x = 0, \tag{37}$$

with $\alpha(x) = \frac{1}{2\sigma^2}\left(\|Ax\|^2 + \mathbb{E}\left[\|A\epsilon\|^2\right] - \|Ay\|^2\right)$.

Let us take a non-zero solution $x$ of $\nabla\mathcal{L} = 0$. Then, the matrix $\alpha(x)I_M + A^T A$ must be singular, which is equivalent to the fact that $-\alpha(x) = \lambda \in \mathrm{sp}(A^T A)$. Being a zero of $\nabla\mathcal{L}$ leads to $A^T Ax = \lambda x$, so that $\|Ax\|^2 = x^T A^T Ax = \lambda\|x\|^2$ and then $\|A^T Ax\|^2 = \lambda\|Ax\|^2$. By injecting this to Eq. (34), we finally get:

$$\mathcal{L}(x) = \mathcal{L}(0) - \|Ax\|^4, \tag{38}$$

$$= \mathcal{L}(0) - \left(\|Ay\|^2 - \mathbb{E}\left[\|A\epsilon\|^2\right] - 2\sigma^2\lambda\right)^2. \tag{39}$$

Therefore, the global minimum is reached when $\lambda$ is the smallest eigenvalue of $A^T A$ constrained by $\|Ax\|^2 = \|Ay\|^2 - \mathbb{E}\left[\|A\epsilon\|^2\right] - 2\sigma^2\lambda \geq 0$. Reciprocally, introducing

$$\Lambda = \{\lambda \in \mathrm{sp}(A^T A) \text{ such that } \|Ay\|^2 - \mathbb{E}\left[\|A\epsilon\|^2\right] - 2\sigma^2\lambda \geq 0\}, \tag{40}$$

we verify that, if $\Lambda = \emptyset$, the global minimizer of $\mathcal{L}$ is 0, and if $\Lambda \neq \emptyset$, the global minimizers are eigenvectors $x$ of $A^T A$ associated with $\min\Lambda$ such that $\|x\|^2 = \frac{1}{\min\Lambda}\left(\|Ay\|^2 - \mathbb{E}\left[\|A\epsilon\|^2\right]\right) - 2\sigma^2$. $\qquad\square$

### A.1.4 $\phi(x) = (\|A_1 x\|^2, \ldots, \|A_K x\|^2)$ with $A_1, \ldots, A_K$ injective and $A_1^T A_1, \ldots, A_K^T A_K$ co-diagonalizable

For any matrix $A$ of size $M \times P$ and a subset $\Xi_Q = \{k_1 < \cdots < k_Q\}$ of $\{1, \ldots, M\}$, we denote by $A_{\Xi_Q}$ the $Q \times P$ submatrix of $A$ such that $[A_{\Xi_Q}]_{i,j} = A_{k_i,j}$. We also denote by $A^+$ the Moore-Penrose inverse of the matrix $A$. Finally, we denote by $\mathrm{span}(e_1, \ldots, e_n)$ the subspace generated by the vectors $e_1, \ldots, e_n$.

**Proposition 2.4.** *We consider $\phi(x) = (\|A_1 x\|^2, \ldots, \|A_K x\|^2)$ with $A_1, \ldots, A_K$ injective and $p(\epsilon_0) \sim \mathcal{N}(0, \sigma^2 I_M)$. Introducing $S_{A_i} = A_i^T A_i$, we assume that $S_{A_1}, \ldots, S_{A_K}$ are co-diagonalizable. We choose an orthonormal basis $(e_1, \ldots, e_M)$ to co-diagonalize them (there always exists one). We call $\Lambda = (\lambda_{i,j})_{1 \leq i \leq M, 1 \leq j \leq K}$ the corresponding matrix of eigenvalues, where $\lambda_{i,j}$ is the eigenvalue of $S_{A_j}$ associated with $e_i$. We also assume that $\Lambda$ is of rank $K$ and that any submatrix of $\Lambda$ with size $K \times K$ is invertible.*

*In that case, introducing:*

$$\Delta = (\Delta_1, \ldots, \Delta_M) \text{ with } \Delta_i = \sum_{j=1}^{K}\left(\lambda_{i,j}(\|A_j y\|^2 - \mathbb{E}\left[\|A_j\epsilon\|^2\right]) - 2\sigma^2\lambda_{i,j}^2\right), \tag{41}$$

$$\Gamma = \{\Xi \subset \{1, \ldots, M\} \mid \forall 1 \leq i \leq K, \left[\Lambda_\Xi^+ \Delta_\Xi\right]_i \geq 0\}, \tag{42}$$

*if $\Gamma = \emptyset$, then the global minimizer of $\mathcal{L}$ is unique equal to 0, otherwise, the minimizers are the $x \in \mathrm{span}(e_i)_{i \in \Xi}$ satisfying $(\|A_1 x\|^2, \ldots, \|A_K x\|^2) = \Lambda_\Xi^+ \Delta_\Xi$ where $\Xi \in \underset{\tilde{\Xi} \in \Gamma}{\mathrm{argmax}}\left\|\Lambda_{\tilde{\Xi}}^+ \Delta_{\tilde{\Xi}}\right\|$.*

*Proof.* We have:

$$\mathcal{L}(x) = \mathbb{E}\left[\sum_{i=1}^{K}\left(\|A_i(x+\epsilon)\|^2 - \|A_i y\|^2\right)^2\right], \tag{43}$$

$$= \sum_{i=1}^{K}\mathbb{E}\left[\|A_i(x+\epsilon)\|^4 + \|A_i y\|^4 - 2\|A_i(x+\epsilon)\|^2\|A_i y\|^2\right]. \tag{44}$$

Using the facts that $\mathbb{E}[A_i\epsilon] = 0$, $\mathbb{E}\left[A_i\epsilon\|A\epsilon\|^2\right] = 0$, and $\mathbb{E}\left[\epsilon\epsilon^T\right] = \sigma^2 I_M$, this loss reads:

$$\mathcal{L}(x) = \sum_{i=1}^{K}\left[\|A_i x\|^4 + 4\sigma^2\|A_i^T A_i x\|^2 + 2\|A_i x\|^2\left(\mathbb{E}\left[\|A_i\epsilon\|^2\right] - \|A_i y\|^2\right)\right] + \mathcal{L}(0). \tag{45}$$

The existence of the global minimum is guaranteed for the same reasons as Prop. 2.3.

The expression of $\nabla\mathcal{L}(x)$ reads:

$$\nabla\mathcal{L}(x) = \sum_{i=1}^{K}\left[4\left(\|A_i x\|^2 + \mathbb{E}\left[\|A_i\epsilon\|^2\right] - \|A_i y\|^2\right)A_i^T A_i x + 8\sigma^2(A_i^T A_i)^2 x\right], \tag{46}$$

and we have:

$$\nabla\mathcal{L}(x) = 0 \iff \left[\sum_{i=1}^{K}\alpha_{A_i}(x)S_{A_i} + S_{A_i}^2\right]x = 0, \tag{47}$$

where we have introduced:

$$\alpha_{A_i}(x) = \left(\|A_i x\|^2 + \mathbb{E}\left[\|A_i\epsilon\|^2\right] - \|A_i y\|^2\right)/(2\sigma^2), \tag{48}$$

$$S_{A_i} = A_i^T A_i. \tag{49}$$

We notice that:

$$\nabla\mathcal{L}(x) = 0 \implies x^T\nabla\mathcal{L}(x) = 0, \tag{50}$$

$$\iff x^T\sum_{i=1}^{N}S_{A_i}^2 x = -\sum_{i=1}^{N}\alpha_{A_i}(x)x^T S_{A_i}x, \tag{51}$$

$$\iff \sum_{i=1}^{N}\|S_{A_i}x\|^2 = -\sum_{i=1}^{N}\alpha_{A_i}(x)\|A_i x\|^2. \tag{52}$$

Therefore, with $x$ a global minimizer of $\mathcal{L}(x)$, we have:

$$\mathcal{L}(x) = \sum_{i=1}^{K}\left[\|A_i x\|^4 - 4\sigma^2\alpha_{A_i}(x)\|A_i x\|^2 + 2\|A_i x\|^2\left(\mathbb{E}\left[\|A_i\epsilon\|^2\right] - \|A_i y\|^2\right)\right] + \mathcal{L}(0), \tag{53}$$

$$= \sum_{i=1}^{K}\|A_i x\|^2\left[\|A_i x\|^2 - 4\sigma^2\alpha_{A_i}(x) + 2\mathbb{E}\left[\|A_i\epsilon\|^2\right] - 2\|A_i y\|^2\right] + \mathcal{L}(0), \tag{54}$$

$$= \mathcal{L}(0) - \sum_{i=1}^{K}\|A_i x\|^4. \tag{55}$$

We take a non-zero global minimizer $x = \sum_{i=1}^{M}x_i e_i$. Introducing $\Xi_Q = \{i \in \{1,\ldots,M\} \mid x_i \neq 0\}$, for all $i \in \Xi_Q$, Eq. (47) leads to:

$$\sum_{j=1}^{K}\alpha_{A_j}(x)\lambda_{i,j} + \lambda_{i,j}^2 = 0, \tag{56}$$

$$\Longleftrightarrow \sum_{j=1}^{K} \lambda_{i,j} \left\|A_j x\right\|^2 = \sum_{j=1}^{K} \left( \lambda_{i,j} (\left\|A_j y\right\|^2 - \mathbb{E}\left[\left\|A_j \epsilon\right\|^2\right]) - 2\sigma^2 \lambda_{i,j}^2 \right), \tag{57}$$

$$\Longleftrightarrow \left[\Lambda N_{Ax}\right]_i = \Delta_i, \tag{58}$$

where we have introduced:

$$N_{Ax} = (\left\|A_1 x\right\|^2, \dots, \left\|A_K x\right\|^2)^T, \tag{59}$$

$$\Delta_i = \sum_{j=1}^{K} \left( \lambda_{i,j} (\left\|A_j y\right\|^2 - \mathbb{E}\left[\left\|A_j \epsilon\right\|^2\right]) - 2\sigma^2 \lambda_{i,j}^2 \right). \tag{60}$$

In a more compact form, we have:

$$\Lambda_{\Xi_Q} N_{Ax} = \Delta_{\Xi_Q}. \tag{61}$$

By assumption, if $Q \geq K$ then the $K \times K$ matrix $\Lambda_{\Xi_Q}^T \Lambda_{\Xi_Q}$ is invertible, and if $Q < K$, then the $Q \times Q$ matrix $\Lambda_{\Xi_Q} \Lambda_{\Xi_Q}^T$ is invertible. Using the fact that $N_{Ax} = \Lambda_{\Xi_Q}^T N_{x,\Xi_Q}$ where $N_x = (x_1^2, \dots, x_M^2)^T$, we show that:

$$N_{Ax} = \begin{cases} (\Lambda_{\Xi_Q}^T \Lambda_{\Xi_Q})^{-1} \Lambda_{\Xi_Q}^T \Delta_{\Xi_Q}, & \text{if } Q \geq K, \\ \Lambda_{\Xi_Q}^T (\Lambda_{\Xi_Q} \Lambda_{\Xi_Q}^T)^{-1} \Delta_{\Xi_Q}, & \text{if } Q < K, \end{cases} \tag{62}$$

that is:

$$N_{Ax} = \Lambda_{\Xi_Q}^+ \Delta_{\Xi_Q}. \tag{63}$$

We note that necessarily, all components of $\Lambda_{\Xi_Q}^+ \Delta_{\Xi_Q}$ are positive.

By injecting this previous expression in $\mathcal{L}(x)$, we get:

$$\mathcal{L}(x) = \mathcal{L}(0) - \left\|N_{Ax}\right\|^2, \tag{64}$$

$$= \mathcal{L}(0) - \left\|\Lambda_{\Xi_Q}^+ \Delta_{\Xi_Q}\right\|^2. \tag{65}$$

Reciprocally, introducing:

$$\Gamma = \{\Xi \subset \{1, \dots, M\} \mid \forall 1 \leq i \leq K, \left[\Lambda_{\Xi}^+ \Delta_{\Xi}\right]_i \geq 0\}, \tag{66}$$

we verify that, if $\Gamma = \emptyset$, the global minimizer of $\mathcal{L}$ is 0, otherwise, the global minimizers are the $x \in \text{span}(e_i)_{i \in \Xi}$ satisfying $N_{Ax} = \Lambda_{\Xi}^+ \Delta_{\Xi}$ where $\Xi \in \underset{\tilde{\Xi} \in \Gamma}{\text{argmax}} \left\|\Lambda_{\tilde{\Xi}}^+ \Delta_{\tilde{\Xi}}\right\|$. $\qquad\square$

For $K = 1$, we check that we recover Prop. 2.3.

## A.2 Proofs of Sect. 4

### A.2.1 Proof of Eq. (14)

**Proposition 4.1.** *For $\phi$ a twice differentiable function, and $\epsilon_0$ arbitrarily distributed with $\mathbb{E}\left[\epsilon_0\right] = 0$, we have:*

$$\mathcal{L}(x, \alpha) = \left\|\phi(x) - \phi(y)\right\|^2 + \alpha^2 \left[ \langle J_\phi(x)^T J_\phi(x), \Sigma \rangle_{\text{F}} + \langle H_\phi(x), \Sigma \rangle_{\text{F}} \cdot (\phi(x) - \phi(y)) \right] + o(\alpha^2), \tag{67}$$

*where $J_\phi(x)$ is the Jacobian matrix of $\phi$ (of size $K \times M$), $H_\phi(x)$ is its Hessian tensor (of rank 3, and size $K \times M \times M$), and $\Sigma$ is the covariance matrix of $\epsilon_0$.*

*Proof.* We introduce $\alpha \in \mathbb{R}$, and assume that $\phi$ is twice differentiable, and that $\epsilon_0$ is arbitrarily distributed with $\mathbb{E}[\epsilon_0] = 0$[10]. In this context, we expand $\phi(x + \alpha\epsilon)$ as a function of $t$ at second order as follows:

$$\phi(x + \alpha\epsilon) = \phi(x) + \alpha J_\phi(x)\epsilon + \frac{1}{2}\alpha^2 \epsilon^T H_\phi(x)\epsilon + o(\alpha^2), \tag{68}$$

where $J_\phi(x) = \left(\frac{\partial \phi_i}{\partial x_j}(x)\right)_{i,j}$ is the Jacobian matrix of $\phi$ (of size $K \times M$), and $H_\phi(x) = \left(\frac{\partial^2 \phi_i}{\partial x_j \partial x_k}(x)\right)_{i,j,k}$ is a Hessian tensor of rank 3 and size $K \times M \times M$. The second order term must be understood as a contraction on the second and third indices of the Hessian tensor.

Let us propagate this expansion in:

$$\mathcal{L}(x, \alpha) = \mathbb{E}_{\epsilon \sim p(\epsilon_0)}\left[\|\phi(x + \alpha\epsilon) - \phi(y)\|^2\right], \tag{69}$$

$$= \mathbb{E}\left[\left\|(\phi(x) - \phi(y)) + \alpha J_\phi(x)\epsilon + \frac{1}{2}\alpha^2 \epsilon^T H_\phi(x)\epsilon + o(\alpha^2)\right\|^2\right], \tag{70}$$

$$= \|(\phi(x) - \phi(y))\|^2 + \alpha^2 \mathbb{E}\left[\|J_\phi(x)\epsilon\|^2\right] + 2\alpha\left(\phi(x) - \phi(y)\right) \cdot \mathbb{E}\left[J_\phi(x)\epsilon\right] \tag{71}$$

$$+ \alpha^2\left(\phi(x) - \phi(y)\right) \cdot \mathbb{E}\left[\epsilon^T H_\phi(x)\epsilon\right] + o(\alpha^2), \tag{72}$$

$$= \|(\phi(x) - \phi(y))\|^2 + \alpha^2\left(\mathbb{E}\left[\|J_\phi(x)\epsilon\|^2\right] + (\phi(x) - \phi(y)) \cdot \mathbb{E}\left[\epsilon^T H_\phi(x)\epsilon\right]\right) + o(\alpha^2), \tag{73}$$

where we have used the fact that $\mathbb{E}[J_\phi(x)\epsilon] = J_\phi(x)\mathbb{E}[\epsilon] = 0$. Introducing the covariance matrix $\Sigma$ of $\epsilon$, we verify that:

$$\mathbb{E}\left[\|J_\phi(x)\epsilon\|^2\right] = \mathbb{E}\left[\epsilon J_\phi(x)^T J_\phi(x)\epsilon\right] = \langle J_\phi(x)^T J_\phi(x), \Sigma\rangle_{\mathrm{F}}, \tag{74}$$

$$\mathbb{E}\left[\epsilon^T H_\phi(x)\epsilon\right] = \langle H_\phi(x), \Sigma\rangle_{\mathrm{F}}, \tag{75}$$

so that:

$$\mathcal{L}(x, \alpha) = \|\phi(x) - \phi(y)\|^2 + \alpha^2\left[\langle J_\phi(x)^T J_\phi(x), \Sigma\rangle_{\mathrm{F}} + \langle H_\phi(x), \Sigma\rangle_{\mathrm{F}} \cdot (\phi(x) - \phi(y))\right] + o(\alpha^2). \tag{76}$$

$$\square$$

## A.3 Proofs of Sect. 4.3

We compute the explicit expressions of the Jacobian matrix $J_\phi(x)$ and the Hessian tensor $H_\phi(x)$ for $\phi$ the operator giving the WPH statistics employed in Sect. 3.2. We break down these computations by first computing first and second-order derivatives for simpler $\phi$ functions, namely $\phi(x) = \psi \star x$ and $\phi(x) = |\psi \star x|$. We assume periodic boundary conditions, so that for any $v \in \mathbb{K}^M$, we formally manipulate $\tilde{v} \in \mathbb{K}^{\mathbb{Z}}$ defined by $\tilde{v}[i] = v[i \bmod M]$ for any $i \in \mathbb{Z}$, and the 'tilde' symbol is omitted for convenience. Below, the filters $\psi$, $\psi_1$, and $\psi_2$ are assumed to be complex-valued filters. For a filter $\psi$, we define the adjoint filter $\psi^\dagger$ by $\psi^\dagger[i] = \overline{\psi}[-i]$. The convolution operation corresponds to the periodic convolution. For $z \in \mathbb{C}^*$, we introduce $\mathrm{sg}(z) = z/|z|$. The notation $\langle x\rangle$ refers to the average over the components of $x$, that is $\langle x\rangle = \frac{1}{Q}\sum_{i=1}^Q x_i$. We denote the element-wise product of $x$ and $y$ by $xy$, or $x \odot y$ when there is ambiguity.

**Derivatives of $\phi(x) = \psi \star x$**  Since $\phi$ is linear, we simply have:

$$J_\phi(x) = \psi \star \cdot, \tag{77}$$

$$H_\phi(x) = 0, \tag{78}$$

that is:

$$\frac{\partial}{\partial x_j}(\psi \star x[i]) = \psi[i - j], \tag{79}$$

$$\frac{\partial^2}{\partial x_j \partial x_k}(\psi \star x[i]) = 0. \tag{80}$$

---

[10]Note that we can always redefine $x_0 \leftarrow x_0 + \mathbb{E}[\epsilon_0]$ for this to be the case.

**Derivatives of $\phi(x) = |\psi \star x|$** Where $\phi(x)$ is nonzero, we have:

$$\frac{\partial}{\partial x_j} \left( |\psi \star x|[i] \right) = \frac{\partial}{\partial x_j} \sqrt{|\psi \star x|^2[i]}, \tag{81}$$

$$= \frac{1}{|\psi \star x|[i]} \left( \mathrm{Re}(\psi \star x)[i] \, \mathrm{Re}(\psi)[i-j] + \mathrm{Im}(\psi \star x)[i] \, \mathrm{Im}(\psi)[i-j] \right), \tag{82}$$

$$= \frac{1}{|\psi \star x|[i]} \mathrm{Re} \left( \psi \star x[i] \overline{\psi}[i-j] \right), \tag{83}$$

$$= \mathrm{Re} \left[ \mathrm{sg}(\psi \star x)[i] \overline{\psi}[i-j] \right], \tag{84}$$

which leads to:

$$J_\phi(x) = \mathrm{Re} \left[ (\mathrm{sg}\, \psi \star x) \odot (\overline{\psi} \star \cdot) \right]. \tag{85}$$

The computation of $H_\phi(x)$ now demands to derive $x \to \mathrm{sg}(\psi \star x)$:

$$\frac{\partial}{\partial x_j} \left( \mathrm{sg}\,(\psi \star x)[i] \right) = \frac{\partial}{\partial x_j} \psi \star x[i] \times \frac{1}{|\psi \star x|[i]} - \frac{\psi \star x[i]}{|\psi \star x|^2[i]} \frac{\partial}{\partial x_j} |\psi \star x|[i], \tag{86}$$

$$= \frac{\psi[i-j]}{|\psi \star x|[i]} - \frac{\psi \star x[i]}{|\psi \star x|^2[i]} \mathrm{Re} \left( \mathrm{sg}(\psi \star x)[i] \overline{\psi}[i-j] \right), \tag{87}$$

$$= \frac{1}{2|\psi \star x|[i]} \left[ \psi[i-j] - \frac{(\psi \star x)^2}{|\psi \star x|^2}[i] \overline{\psi}[i-j] \right], \tag{88}$$

$$= \frac{1}{2|\psi \star x|[i]} \left[ \psi[i-j] - \mathrm{sg}(\psi \star x)^2[i] \overline{\psi}[i-j] \right]. \tag{89}$$

Therefore:

$$\frac{\partial^2}{\partial x_j \partial x_k} \left( |\psi \star x|[i] \right) = \mathrm{Re} \left[ \frac{\partial}{\partial x_k} \left( \mathrm{sg}(\psi \star x)[i] \right) \overline{\psi}[i-j] \right], \tag{90}$$

$$= \frac{1}{2|\psi \star x|[i]} \mathrm{Re} \left[ \psi[i-k] \overline{\psi}[i-j] - \mathrm{sg}(\psi \star x)^2[i] \overline{\psi}[i-k] \overline{\psi}[i-j] \right]. \tag{91}$$

**Derivatives of $\phi(x) = \langle |\psi \star x|^2 \rangle$** Using the above, we get:

$$\frac{\partial}{\partial x_j} \langle |\psi \star x|^2 \rangle = \langle \frac{\partial}{\partial x_j} (\psi \star x) \times \overline{\psi \star x} \rangle + \overline{\langle \frac{\partial}{\partial x_j} (\psi \star x) \times \overline{\psi \star x} \rangle}, \tag{92}$$

$$= \frac{2}{M} \mathrm{Re} \left( \sum_{i=1}^{M} \psi[i-j] \times \overline{\psi \star x}[i] \right), \tag{93}$$

$$= \frac{2}{M} \mathrm{Re} \left( \psi^\dagger \star \psi \star x \right) [j], \tag{94}$$

and:

$$\frac{\partial^2}{\partial x_j \partial x_k} \langle |\psi \star x|^2 \rangle = \frac{2}{M} \mathrm{Re} \left( \psi^\dagger \star \psi \right) [j-k]. \tag{95}$$

**Derivatives of $\phi(x) = \langle |\psi \star x| \rangle^2$** We get:

$$\frac{\partial}{\partial x_j} \langle |\psi \star x| \rangle^2 = 2 \langle |\psi \star x| \rangle \times \frac{1}{M} \sum_{i=1}^{M} \mathrm{Re} \left[ \mathrm{sg}(\psi \star x)[i] \overline{\psi}[i-j] \right], \tag{96}$$

$$= \frac{2}{M} \langle |\psi \star x| \rangle \, \mathrm{Re} \left[ \psi^\dagger \star \mathrm{sg}(\psi \star x) \right] [j], \tag{97}$$

and:

$$\frac{\partial^2}{\partial x_j \partial x_k} \langle |\psi \star x| \rangle^2 = 2 \frac{\partial}{\partial x_j} \langle |\psi \star x| \rangle \frac{\partial}{\partial x_k} \langle |\psi \star x| \rangle + 2 \langle |\psi \star x| \rangle \frac{\partial^2}{\partial x_j \partial x_k} \langle |\psi \star x| \rangle, \tag{98}$$

$$= \frac{2}{M^2} \operatorname{Re}\left[\psi^\dagger \star \operatorname{sg}(\psi \star x)\right][j] \times \operatorname{Re}\left[\psi^\dagger \star \operatorname{sg}(\psi \star x)\right][k] \tag{99}$$

$$+ \frac{1}{M}\langle|\psi \star x|\rangle \sum_{i=1}^{M}\operatorname{Re}\left[\frac{\psi^\dagger[j-i]\overline{\psi}^\dagger[k-i]}{|\psi \star x|[i]} - \frac{\psi^\dagger[j-i]\operatorname{sg}(\psi \star x)^2[i]\psi^\dagger[k-i]}{|\psi \star x|[i]}\right], \tag{100}$$

$$= \frac{2}{M^2} \operatorname{Re}\left[\psi^\dagger \star \operatorname{sg}(\psi \star x)\right][j] \times \operatorname{Re}\left[\psi^\dagger \star \operatorname{sg}(\psi \star x)\right][k] \tag{101}$$

$$+ \frac{1}{M}\langle|\psi \star x|\rangle \times \operatorname{Re}\left[\psi^\dagger \otimes \frac{1}{|\psi \star x|} \otimes \overline{\psi}^\dagger[j,k] - \psi^\dagger \otimes \frac{\operatorname{sg}(\psi \star x)^2}{|\psi \star x|} \otimes \psi^\dagger[j,k]\right], \tag{102}$$

where we have introduced the notation $a \otimes b \otimes c\ [j,k] = \sum_{i=1}^{M} a[j-i]b[i]c[k-i]$. Conveniently, note that $a \otimes b \otimes c\ [j,j] = ac \star b[j]$.

**Derivatives of $\phi(x) = \langle|\psi_1 \star x| \times \overline{\psi_2 \star x}\rangle$**   We get:

$$\frac{\partial}{\partial x_j}\langle|\psi_1 \star x|\overline{\psi_2 \star x}\rangle = \langle\frac{\partial}{\partial x_j}\left(|\psi_1 \star x|\right) \times \overline{\psi_2} \star x\rangle + \langle|\psi_1 \star x| \times \frac{\partial}{\partial x_j}\left(\overline{\psi_2} \star x\right)\rangle, \tag{103}$$

$$= \frac{1}{M}\sum_{i=1}^{M}\left[\operatorname{Re}\left[\operatorname{sg}(\psi_1 \star x)[i]\overline{\psi_1}[i-j]\right]\overline{\psi_2} \star x[i] + |\psi_1 \star x|[i]\overline{\psi_2}[i-j]\right], \tag{104}$$

$$= \frac{1}{2M}\sum_{i=1}^{M}\left[\operatorname{sg}(\psi_1 \star x)[i]\overline{\psi_1}[i-j]\overline{\psi_2} \star x[i] + \overline{\operatorname{sg}(\psi_1 \star x)}[i]\psi_1[i-j]\overline{\psi_2} \star x[i]\right] \tag{105}$$

$$+ \frac{1}{M}\psi_2^\dagger \star |\psi_1 \star x|[j], \tag{106}$$

$$= \frac{1}{2M}\left[\psi_1^\dagger \star \left(\operatorname{sg}(\psi_1 \star x) \times \overline{\psi_2} \star x\right) + \overline{\psi_1^\dagger \star \left(\operatorname{sg}(\psi_1 \star x) \times \psi_2 \star x\right)}\right][j] \tag{107}$$

$$+ \frac{1}{M}\psi_2^\dagger \star |\psi_1 \star x|[j]. \tag{108}$$

We break down the calculation of the second-order derivatives of $\phi$ by first calculating:

$$\frac{\partial}{\partial x_k}\left(\psi_2^\dagger \star |\psi_1 \star x|[j]\right) = \sum_{i=1}^{M}\psi_2^\dagger[j-i]\operatorname{Re}\left[\operatorname{sg}(\psi_1 \star x)[i]\overline{\psi_1}[i-k]\right], \tag{109}$$

$$= \frac{1}{2}\sum_{i=1}^{M}\left[\psi_1^\dagger[k-i]\operatorname{sg}(\psi_1 \star x)[i]\psi_2^\dagger[j-i]\right. \tag{110}$$

$$\left.+\overline{\psi}_1^\dagger[k-i]\overline{\operatorname{sg}(\psi_1 \star x)}[i]\psi_2^\dagger[j-i]\right], \tag{111}$$

$$= \frac{1}{2}\left[\psi_2^\dagger \otimes \operatorname{sg}(\psi_1 \star x) \otimes \psi_1^\dagger + \psi_2^\dagger \otimes \overline{\operatorname{sg}(\psi_1 \star x)} \otimes \overline{\psi_1}^\dagger\right][j,k]. \tag{112}$$

We also have:

$$\frac{\partial}{\partial x_k}\left(\operatorname{sg}(\psi_1 \star x) \times \psi_2 \star x\right)[i] = \frac{\psi_2 \star x[i]}{2|\psi_1 \star x|[i]}\left[\overline{\psi_1}^\dagger[k-i] - \operatorname{sg}(\psi_1 \star x)^2[i]\psi_1^\dagger[k-i]\right] \tag{113}$$

$$+ \operatorname{sg}(\psi_1 \star x)[i]\overline{\psi_2}^\dagger[k-i], \tag{114}$$

$$\tag{115}$$

so that:

$$\frac{\partial}{\partial x_k}\psi_1^\dagger \star \left(\operatorname{sg}(\psi_1 \star x) \times \psi_2 \star x\right)[j] = \frac{1}{2}\sum_{i=1}^{M}\psi_1^\dagger[j-i]\frac{\psi_2 \star x[i]}{|\psi_1 \star x|[i]}\overline{\psi_1}^\dagger[k-i] \tag{116}$$

$$- \frac{1}{2}\sum_{i=1}^{M}\psi_1^\dagger[j-i]\frac{\psi_2 \star x \times \operatorname{sg}(\psi_1 \star x)^2[i]}{|\psi_1 \star x|[i]}\psi_1^\dagger[k-i] \tag{117}$$

$$+ \sum_{i=1}^{M} \psi_1^\dagger[j-i]\,\mathrm{sg}(\psi_1 \star x)[i]\overline{\psi_2}^\dagger[k-i], \tag{118}$$

$$= \frac{1}{2}\left[\psi_1^\dagger \otimes \frac{\psi_2 \star x}{|\psi_1 \star x|} \otimes \overline{\psi_1}^\dagger + \psi_1^\dagger \otimes \frac{\psi_2 \star x \times \mathrm{sg}(\psi_1 \star x)^2}{|\psi_1 \star x|} \otimes \psi_1^\dagger \right. \tag{119}$$

$$\left. +2\psi_1^\dagger \otimes \mathrm{sg}(\psi_1 \star x) \otimes \overline{\psi_2}^\dagger\right][j,k]. \tag{120}$$

Therefore:

$$\frac{\partial^2}{\partial x_j \partial x_k}\langle|\psi_1 \star x|\overline{\psi_2 \star x}\rangle = \frac{1}{4M}\left[\psi_1^\dagger \otimes \frac{\overline{\psi_2} \star x}{|\psi_1 \star x|} \otimes \overline{\psi_1}^\dagger + \overline{\psi_1}^\dagger \otimes \frac{\overline{\psi_2} \star x}{|\psi_1 \star x|} \otimes \psi_1^\dagger \right. \tag{121}$$

$$- \psi_1^\dagger \otimes \frac{\overline{\psi_2} \star x \times \mathrm{sg}(\psi_1 \star x)^2}{|\psi_1 \star x|} \otimes \psi_1^\dagger - \psi_1^\dagger \otimes \overline{\frac{\psi_2 \star x \times \mathrm{sg}(\psi_1 \star x)^2}{|\psi_1 \star x|}} \otimes \psi_1^\dagger \tag{122}$$

$$+ 2\psi_1^\dagger \otimes \mathrm{sg}(\psi_1 \star x) \otimes \psi_2^\dagger + 2\overline{\psi_1}^\dagger \otimes \overline{\mathrm{sg}(\psi_1 \star x)} \otimes \psi_2^\dagger \tag{123}$$

$$\left. +2\psi_2^\dagger \otimes \mathrm{sg}(\psi_1 \star x) \otimes \psi_1^\dagger + 2\psi_2^\dagger \otimes \overline{\mathrm{sg}(\psi_1 \star x)} \otimes \overline{\psi_1}^\dagger\right][j,k]. \tag{124}$$

**Summary for $\psi^\dagger = \psi$ and $\Sigma = \mathrm{diag}(\sigma^2)$**   For $\phi(x) \in \mathbb{C}$, we only focus on simplified expressions of:

$$\langle J_\phi^\dagger(x)J_\phi(x), \Sigma\rangle_{\mathrm{F}} = \sum_{i=1}^{M}|\frac{\partial \phi}{\partial x_i}(x)|^2 \sigma_i^2, \tag{125}$$

$$\langle H_\phi(x), \Sigma\rangle_{\mathrm{F}} = \sum_{i=1}^{M}\frac{\partial^2 \phi}{\partial x_i^2}(x)\sigma_i^2. \tag{126}$$

We get for $\langle J_\phi^\dagger(x)J_\phi(x), \Sigma\rangle_{\mathrm{F}}$:

$$\phi(x) = \hat{S}^{11}(x) \qquad \rightarrow \frac{4}{M^2}\sum_{i=1}^{M}\mathrm{Re}\,(\psi \star \psi \star x)^2\,[i] \times \sigma^2[i] \tag{127}$$

$$\phi(x) = \hat{S}^{00}(x) \qquad \rightarrow \frac{4}{M^2}\sum_{i=1}^{M}\mathrm{Re}\,[\psi \star \psi \star x - \langle|\psi \star x|\rangle \times \psi \star \mathrm{sg}(\psi \star x)]^2\,[i] \times \sigma^2[i] \tag{128}$$

$$\phi(x) = \hat{S}^{01}(x) \qquad \rightarrow \frac{1}{4M^2}\sum_{i=1}^{M}\left|3\psi \star |\psi \star x| + \overline{\psi \star (\mathrm{sg}(\psi \star x) \times \psi \star x)}\right|^2\,[i] \times \sigma^2[i] \tag{129}$$

$$\phi(x) = \hat{C}^{01}(x) \qquad \rightarrow \frac{1}{4M^2}\sum_{i=1}^{M}\left|\psi_1 \star \left(\mathrm{sg}(\psi_1 \star x) \times \overline{\psi_2} \star x\right) + \overline{\psi_1 \star (\mathrm{sg}(\psi_1 \star x) \times \psi_2 \star x)}\right.\tag{130}$$

$$\left.+2\psi_2 \star |\psi_1 \star x|\right|^2\,[i] \times \sigma^2[i]. \tag{131}$$

And we get for $\langle H_\phi(x), \Sigma\rangle_{\mathrm{F}}$:

$$\phi(x) = \hat{S}^{11}(x) \quad \rightarrow \frac{2}{M}\,\mathrm{Re}\,(\psi \star \psi)\,[0]\sum_{i=1}^{M}\sigma^2[i] \tag{132}$$

$$\phi(x) = \hat{S}^{00}(x) \quad \rightarrow \frac{1}{M}\sum_{i=1}^{M}\mathrm{Re}\left[2\psi \star \psi[0] - \frac{2}{M}\left[\mathrm{Re}\,(\psi \star \mathrm{sg}(\psi \star x))\right]^2 \right. \tag{133}$$

$$\left. +\langle|\psi \star x|\rangle\left(\psi^2 \star \frac{(\mathrm{sg}(\psi \star x))^2}{|\psi \star x|} - |\psi|^2 \star \frac{1}{|\psi \star x|}\right)\right]\,[i] \times \sigma^2[i] \tag{134}$$

$$\phi(x) = \hat{S}^{01}(x) \quad \rightarrow \frac{1}{4M}\sum_{i=1}^{M}\left[6|\psi|^2 \star \overline{\mathrm{sg}(\psi \star x)} + 3\psi^2 \star \mathrm{sg}(\psi \star x)\right. \tag{135}$$

$$-\overline{\psi^2 \star (\text{sg}(\psi \star x)^3)}\Big][i] \times \sigma^2[i] \tag{136}$$

$$\phi(x) = \hat{C}^{01}(x) \quad \rightarrow \frac{1}{4M} \sum_{i=1}^{M} \left[ 2|\psi_1|^2 \star \frac{\overline{\psi_2 \star x}}{|\psi_1 \star x|} + 4\psi_1\psi_2 \star \text{sg}(\psi_1 \star x) + 4\overline{\psi_1\overline{\psi_2} \star \text{sg}(\psi_1 \star x)} \right. \tag{137}$$

$$\left. -\psi_1^2 \star \left( \frac{\overline{\psi_2 \star x}}{|\psi_1 \star x|} \times \text{sg}(\psi_1 \star x)^2 \right) - \overline{\psi_1^2 \star \left( \frac{\psi_2 \star x}{|\psi_1 \star x|} \times \text{sg}(\psi_1 \star x)^2 \right)} \right] [i] \times \sigma^2[i]. \tag{138}$$

## B  Relation to Maximum Likelihood Estimation

We investigate the links between statistical component separation methods and maximum likelihood estimation (MLE).

**Estimation of $x_0$.**  In a denoising context, the goal is to recover $x_0$ given $y$. A naive application of MLE would be to maximize the likelihood $p(y \,|\, x_0)$. Assuming a Gaussian noise with covariance $\Sigma$, we have $p(y \,|\, x_0) \propto \exp\left(-\frac{1}{2}\|y - x_0\|_\Sigma^2\right)$ with $\|y - x_0\|_\Sigma^2 = (y - x_0)^T \Sigma^{-1} (y - x_0)$, which leads to the trivial estimate $\hat{x}_0 = y$. Since in this setting, naive MLE can only bring us back to the noisy data, one typically confuses MLE with the estimation of a maximum a posteriori (MAP), which requires the definition of a prior distribution $p(x)$ over the target signal (see e.g. Elad et al., 2023). With $p(x) \propto \exp(-\lambda\rho(x))$, still in the case of a Gaussian noise, a MAP estimate $\hat{x}_0$ of $x_0$ reads:

$$\hat{x}_0 \in \arg \min_x \left[ \frac{1}{2}\|y - x\|_\Sigma^2 + \lambda\rho(x) \right]. \tag{139}$$

This minimization problem aims to strike a balance between two constraints: one enforcing proximity to the noisy data $y$, and the other imposing a prior constraint, typically in the form of a regularity constraint over $x$. The parameter $\lambda$ controls the relative weight of these two constraints. A statistical component separation method can be remotely related to MLE in this picture. The estimate $\hat{x}_0$ as defined in Eq. (1) can be interpreted as an MLE estimate by choosing $\rho(x) = \mathcal{L}(x) = \mathbb{E}_{\epsilon \sim p(\epsilon_0)}\left[\|\phi(x + \epsilon) - \phi(y)\|_2^2\right]$ and picking $\lambda \to +\infty$. Moreover, by initializing the optimization of $\mathcal{L}(x)$ with $y$, an implicit notion of proximity to $y$ is implied.

**Estimation of $\phi(x_0)$.**  When focusing on the estimation of $\phi(x_0)$ given $y$, which is the ultimate goal of a statistical component separation method, MLE methods require an explicit model connecting $\phi(x_0)$ to $y$. The approach taken in this paper share similarities with the sampling process of maximum entropy models as defined in Bruna & Mallat (2019). For the purpose of this discussion, we model $x_0$ as a realization of a macrocanonical model with density $p(x) = \mathcal{Z}^{-1}\exp\left(-\theta_{\phi(x_0)} \cdot \phi(x)\right)$, where $\theta_{\phi(x_0)}$ is a vector of parameters fixed so that $\mathbb{E}_{x \sim p(x)}[\phi(x)] = \phi(x_0)$. In this context, assuming a Gaussian noise with covariance $\Sigma$, the likelihood $p(y \,|\, \phi(x_0))$ can be written as:

$$p(y \,|\, \phi(x_0)) \propto \int \exp\left(-\theta_{\phi(x_0)} \cdot \phi(x') - \frac{1}{2}\|y - x'\|_\Sigma^2\right) dx'. \tag{140}$$

This likelihood remains generally intractable, which prevents further connections to the statistical component separation method presented in this paper.

## C  Complementary Details on the Data

We give further details on the nature of the data used for the experiments in Sect. 3 and 4.

**The dust image**  This image corresponds to a simulated intensity map of the emission of dust grains in the interstellar medium. It was taken from the work of Régaldo-Saint Blancard et al. (2023), and we refer to this paper for further details on the way it was generated.

**The LSS image** This image was built from a cosmological simulation taken from the `Quijote` suite (Villaescusa-Navarro et al., 2020). This simulation describes the evolution of fluid of dark matter in a $1$ $(\mathrm{Gpc}/h)^3$ periodic box for a $\Lambda$CDM cosmology parameterized by $(\Omega_m, \Omega_b, h, n_s, \sigma_8) = (0.3223, 0.04625, 0.7015, 0.9607, 0.8311)$ (high-resolution LH simulation). We take a snapshot at $z = 0$ of the dark matter density field at $z = 0$ and project a $500 \times 500 \times 10$ $(\mathrm{Mpc}/h)^3$ slice along the thinnest dimension. Our test image is the logarithm of this projection.

**The ImageNet image** We have randomly picked a RGB image of the ImageNet dataset (Deng et al., 2009) from the "digital clock" class. For the experiments of Sect. 3.2, to simplify, we have first preprocessed the image by averaging it over the channel dimension.

## D   Definition and Computation of the WPH statistics

We use the same bump-steerable wavelets as in Regaldo-Saint Blancard et al. (2021) with $J = 7$ and $L = 4$, which leads to a bank of $J \times L = 28$ wavelets. For a given image $x$, the covariances introduced in Sect. 3.2 are estimated as follows:

$$\hat{S}_i^{11}(x) = \langle |x \star \psi_i|^2 \rangle, \qquad \hat{S}_i^{00}(x) = \langle |x \star \psi_i|^2 \rangle - \langle |x \star \psi_i| \rangle^2, \tag{141}$$

$$\hat{S}_i^{01}(x) = \langle |x \star \psi_i| \times \overline{x \star \psi_i} \rangle, \qquad \hat{C}_{i,j}^{01}(x) = \langle |x \star \psi_i| \times \overline{x \star \psi_j} \rangle, \tag{142}$$

where $\langle \cdot \rangle$ is a spatial average operator. Moreoever, in the numerical experiments of this paper, these coefficients are systematically normalized according to the $\hat{S}^{11}$ coefficients of the noisy map $y$ as follows:

$$\tilde{S}_i^{11}(x) = \frac{\hat{S}_i^{11}(x)}{\hat{S}_i^{11}(y)}, \qquad \tilde{S}_i^{00}(x) = \frac{\hat{S}_i^{00}(x)}{\hat{S}_i^{11}(y)}, \tag{143}$$

$$\tilde{S}_i^{01}(x) = \frac{\hat{S}_i^{01}(x)}{\hat{S}_i^{11}(y)}, \qquad \tilde{C}_{i,j}^{01}(x) = \frac{\hat{C}_{i,j}^{01}(x)}{\sqrt{\hat{S}_i^{11}(y)\hat{S}_j^{11}(y)}}. \tag{144}$$

As it was reported in the related literature (Zhang & Mallat, 2021; Allys et al., 2020; Regaldo-Saint Blancard et al., 2021; Régaldo-Saint Blancard et al., 2023), this normalization acts as a preconditioning of the loss function $\mathcal{L}$, and has then a direct impact on the optimum $\hat{x}_0$ one gets with a given optimizer. The quality of the results then a priori depends on this normalization, and it worth noticing than alternative approaches as in Delouis et al. (2022) could have been explored.

## E   Connection with Delouis et al. (2022)

We draw connections between Algorithm 2 and the algorithm introduced in Delouis et al. (2022), which is formally described in Algorithm 3. These connections show that the two algorithms are very related. A further exploration of the pros and cons of each of these two algorithms is left for future work.

The following lemma highlights the similarities between the loss function $\mathcal{L}$ defined in Eq. (1) and the one involved in Algorithm 3.

**Lemma E.1.** *Introducing* $\sigma_\phi(x+\epsilon)^2 = \mathbb{E}\left[|\phi(x+\epsilon)|^2\right] - |\mathbb{E}\left[\phi(x+\epsilon)\right]|^2$, *that is the vector of variances of each component of* $\phi(x+\epsilon)$, *we have:*

$$\mathcal{L}(x) = \mathbb{E}_{\epsilon \sim p(\epsilon_0)}\left[\|\phi(x+\epsilon) - \phi(y)\|^2\right], \tag{145}$$

$$= \|\sigma_\phi(x+\epsilon)\|^2 + \|\mathbb{E}\left[\phi(x+\epsilon)\right] - \phi(y)\|^2. \tag{146}$$

---

**Algorithm 3** Statistical Component Separation as in Delouis et al. (2022)

---

**Input:** $y$, $p(\epsilon_0)$, $Q$, $T$, $P$
**Initialize:** $\hat{x}_0 = y$
**for** $i = 1 \ldots P$ **do**
    **sample** $\epsilon_1, \ldots, \epsilon_Q \sim p(\epsilon_0)$
    $m = \sum_{k=1}^{Q} \phi(\hat{x}_0 + \epsilon_k)/Q$
    $\sigma = \left( \sum_{k=1}^{Q} (\phi(\hat{x}_0 + \epsilon_k) - m)^2/Q \right)^{1/2}$
    $B = m - \phi(\hat{x}_0)$
    **for** $j = 1 \ldots T$ **do**
        $\hat{\mathcal{L}}(\hat{x}_0) = \left\| \left[ \phi(\hat{x}_0) + B - \phi(y) \right] / \sigma \right\|^2$
        $\hat{x}_0 \leftarrow \text{ONE\_STEP\_OPTIM} \left[ \hat{\mathcal{L}}(\hat{x}_0) \right]$
    **end for**
**end for**
**return** $\hat{x}_0$

---

*Proof.*

$$\mathcal{L}(x) = \mathbb{E}_{\epsilon \sim p(\epsilon_0)} \left[ \| \phi(x + \epsilon) - \phi(y) \|^2 \right], \tag{147}$$

$$= \mathbb{E} \left[ \| \phi(x + \epsilon) \|^2 \right] + \| \phi(y) \|^2 - 2\mathbb{E} \left[ \text{Re} \left( \phi(x + \epsilon) \cdot \overline{\phi(y)} \right) \right], \tag{148}$$

$$= \mathbb{E} \left[ \| \phi(x + \epsilon) \|^2 \right] + \| \phi(y) \|^2 - 2 \, \text{Re} \left( \mathbb{E} \left[ \phi(x + \epsilon) \right] \cdot \overline{\phi(y)} \right), \tag{149}$$

$$= \mathbb{E} \left[ \| \phi(x + \epsilon) \|^2 \right] - \| \mathbb{E} \left[ \phi(x + \epsilon) \right] \|^2 + \| \mathbb{E} \left[ \phi(x + \epsilon) \right] \|^2 \tag{150}$$

$$+ \| \phi(y) \|^2 - 2 \, \text{Re} \left( \mathbb{E} \left[ \phi(x + \epsilon) \right] \cdot \overline{\phi(y)} \right), \tag{151}$$

$$= \| \sigma_\phi(x + \epsilon) \|^2 + \| \mathbb{E} \left[ \phi(x + \epsilon) \right] - \phi(y) \|^2. \tag{152}$$

$\square$

Thanks to this lemma, now $\mathcal{L}$ appears as the sum of two terms. The first one constrains the norm of the variance vector $\sigma_\phi(x + \epsilon)$ to be minimal, while the second one constrains the mean vector $\mathbb{E}\left[\phi(x + \epsilon)\right]$ to be close to $\phi(y)$. In the light of this new expression of $\mathcal{L}$, Algorithm 3 aims to minimize the related following loss:

$$\tilde{\mathcal{L}}(x) = \left\| \frac{\mathbb{E} \left[ \phi(x + \epsilon) \right] - \phi(y)}{\sigma_\phi(x + \epsilon)} \right\|^2, \tag{153}$$

which appears to be the second term of Eq. (146) normalized by the first term of the same equation. However, instead of minimizing directly $\tilde{\mathcal{L}}$, Algorithm 2 makes the following approximations:

$$\mathbb{E} \left[ \phi(x + \epsilon) \right] \approx \phi(x) + B, \tag{154}$$

$$\sigma_\phi(x + \epsilon) \approx \sigma, \tag{155}$$

where $B$ and $\sigma$ are repectively the bias and standard deviation terms explicited in Algorithm 2. Same as in Algorithm 2, Algorithm 3 adopts a stepwise approach, where $B$ and $\sigma$ are updated at each step $i \in \{1, \ldots, P\}$ using the $\hat{x}_0$ signal obtained from the previous step.

# F   Additional Results

We show in Figs. F.1, F.2, and F.3 (respectively, F.4, F.5, F.6, and F.7) equivalent results to Figs. 2, 3, and 4, respectively, for the LSS (ImageNet) data. We also show in Fig. F.8 equivalent results to Fig. 2 for the ImageNet data and a ConvNet-based representation.

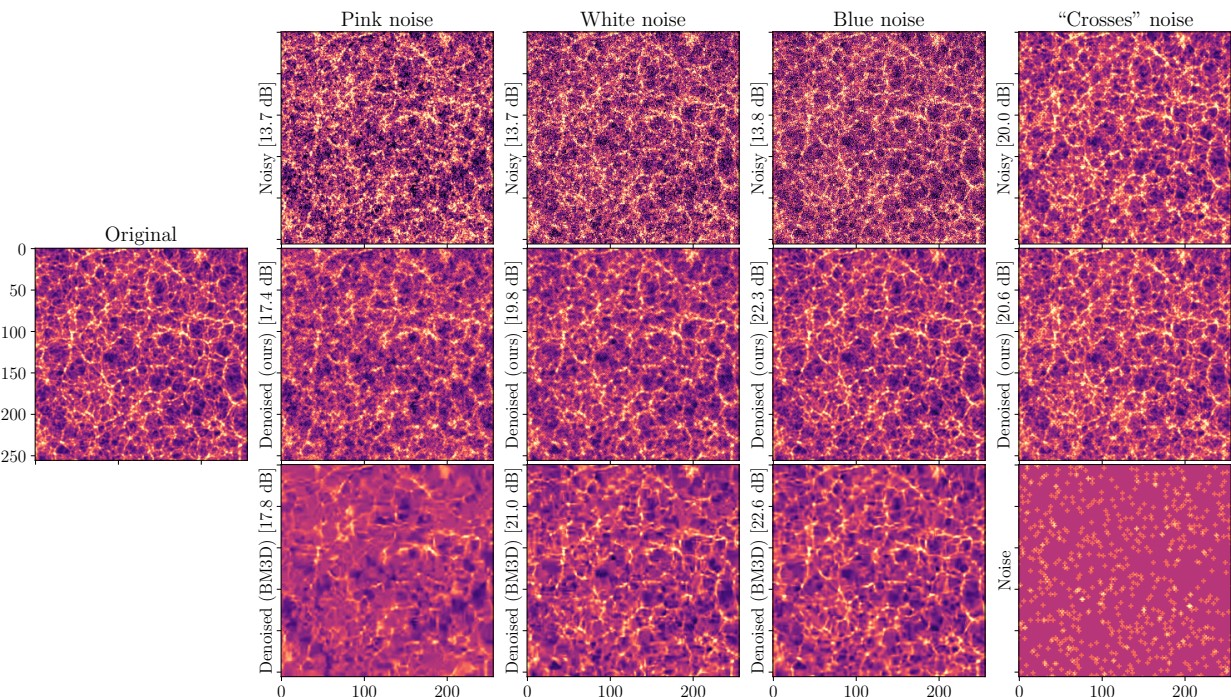

Figure F.1: Same as Fig. 2 for the LSS image.

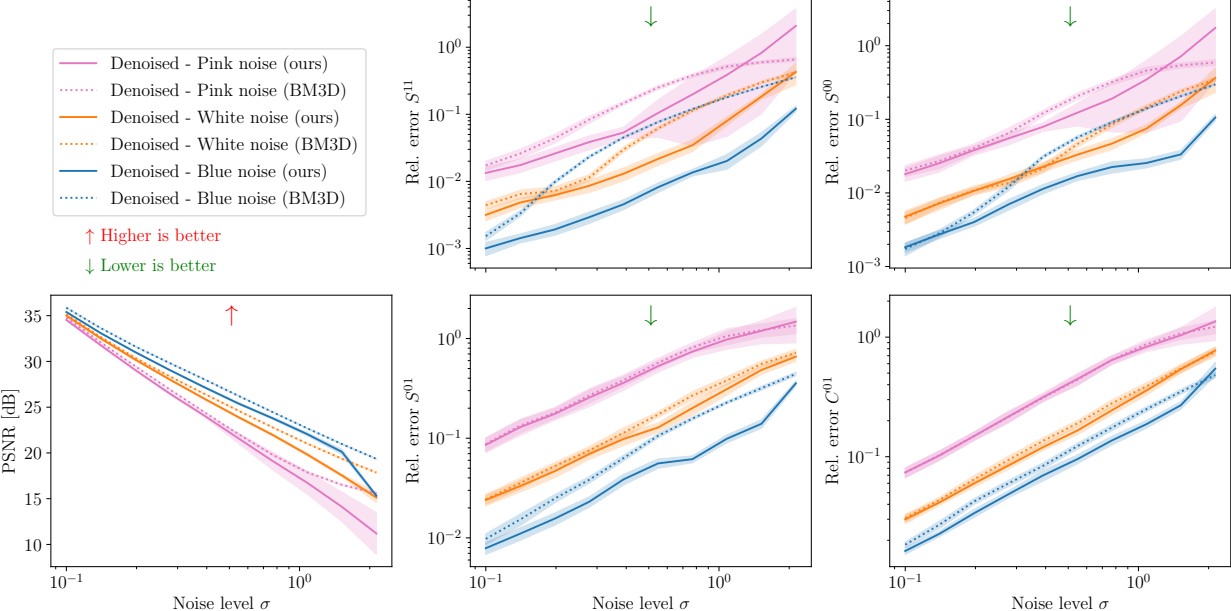

Figure F.2: Same as Fig. 3 for the LSS image.

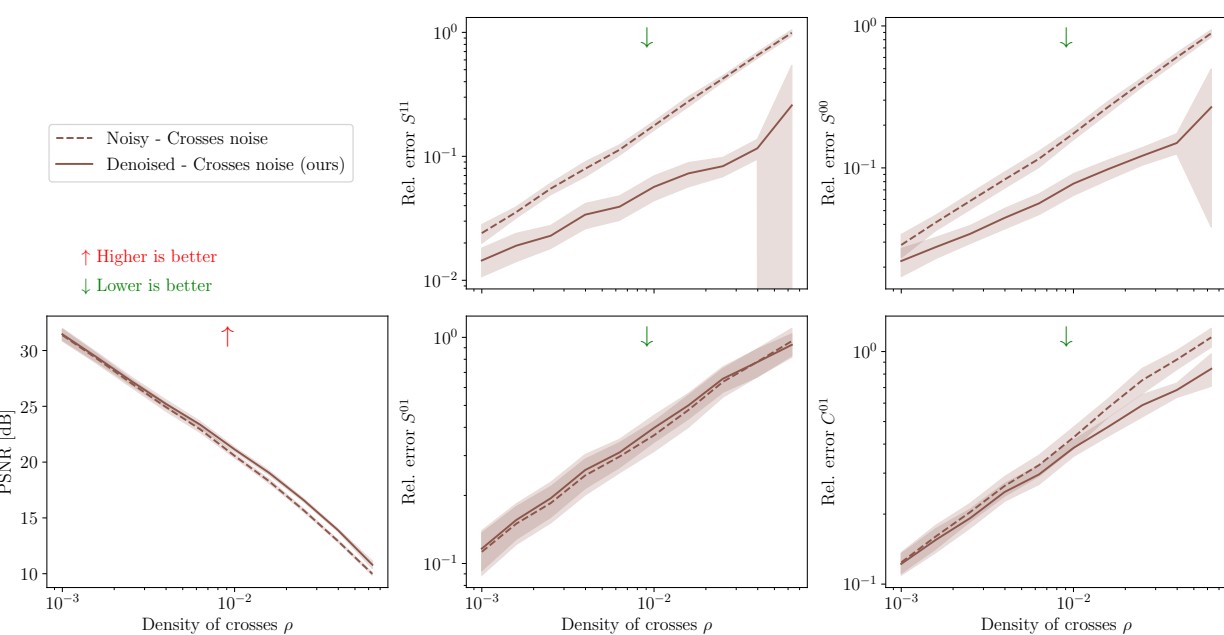

Figure F.3: Same as Fig. 4 for the LSS image.

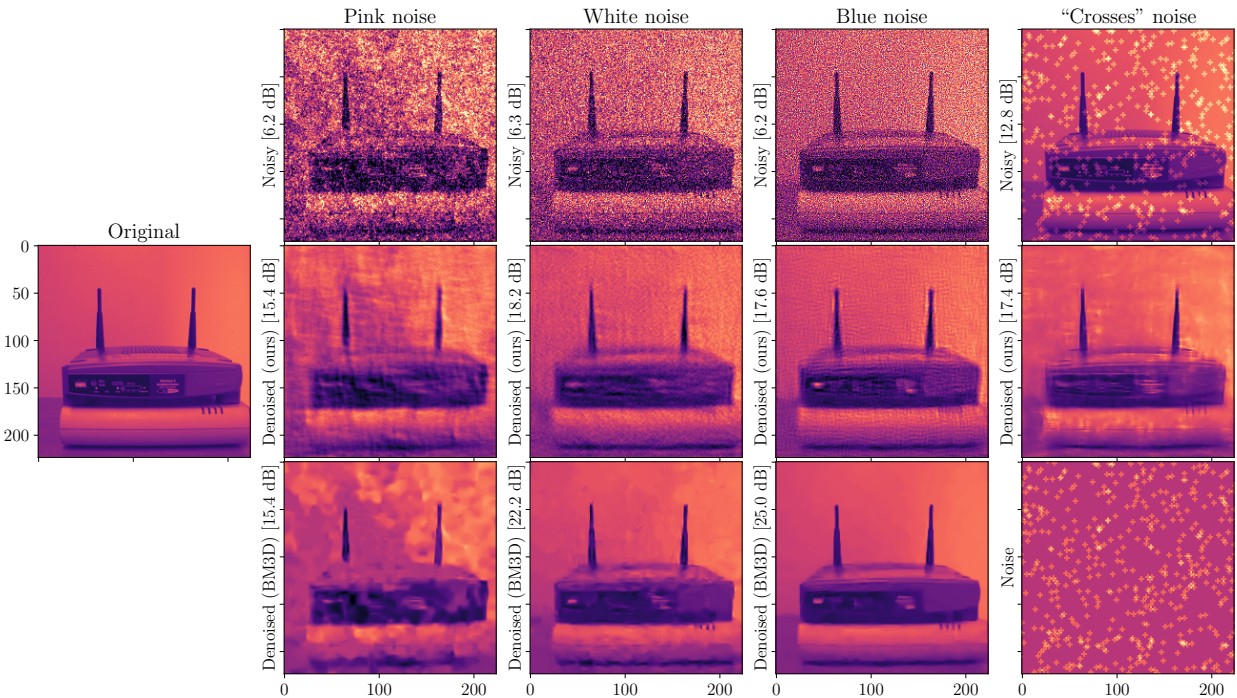

Figure F.4: Same as Fig. 2 for an ImageNet image.

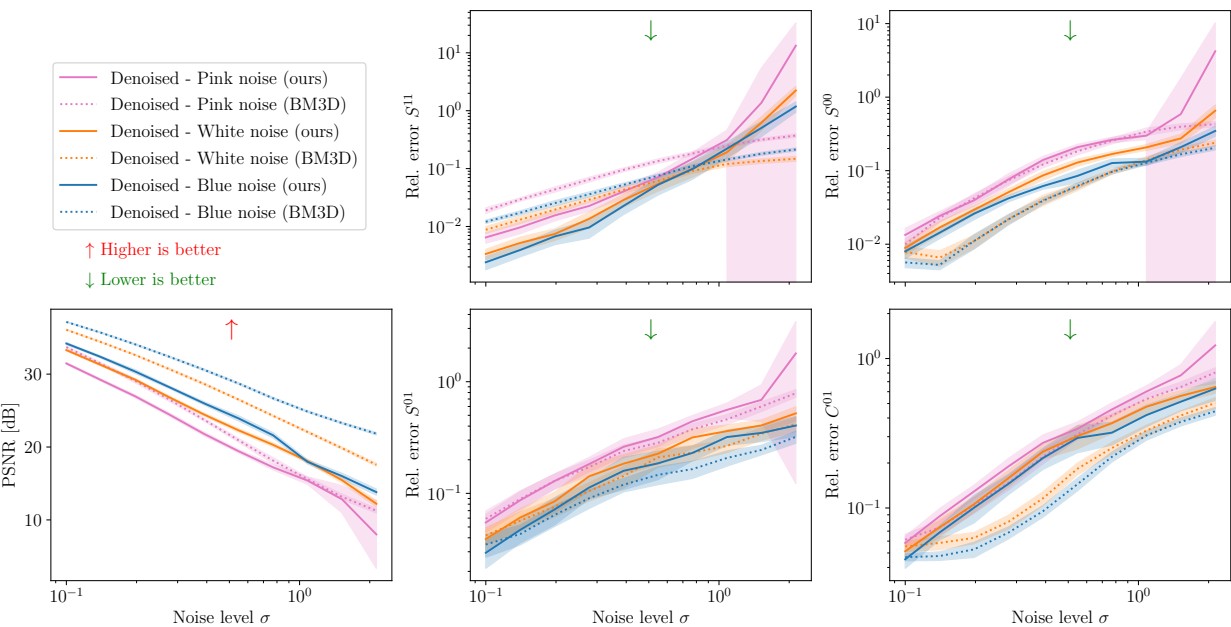

Figure F.5: Same as Fig. 3 for the ImageNet image shown in Fig. F.4.

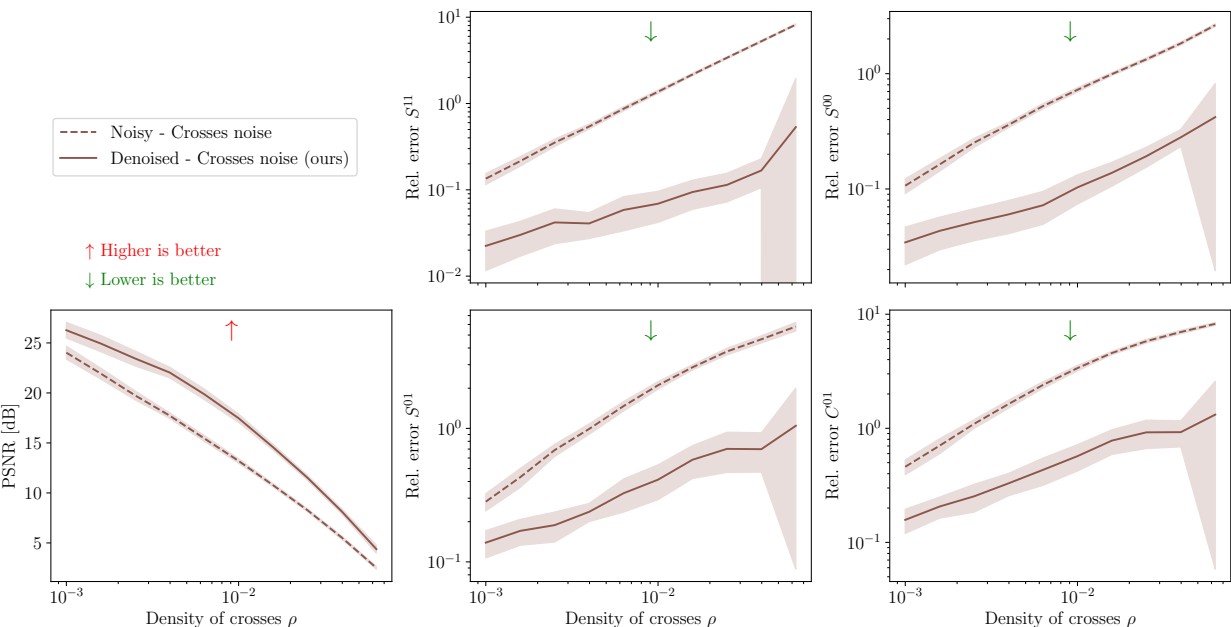

Figure F.6: Same as Fig. 4 for the ImageNet image.

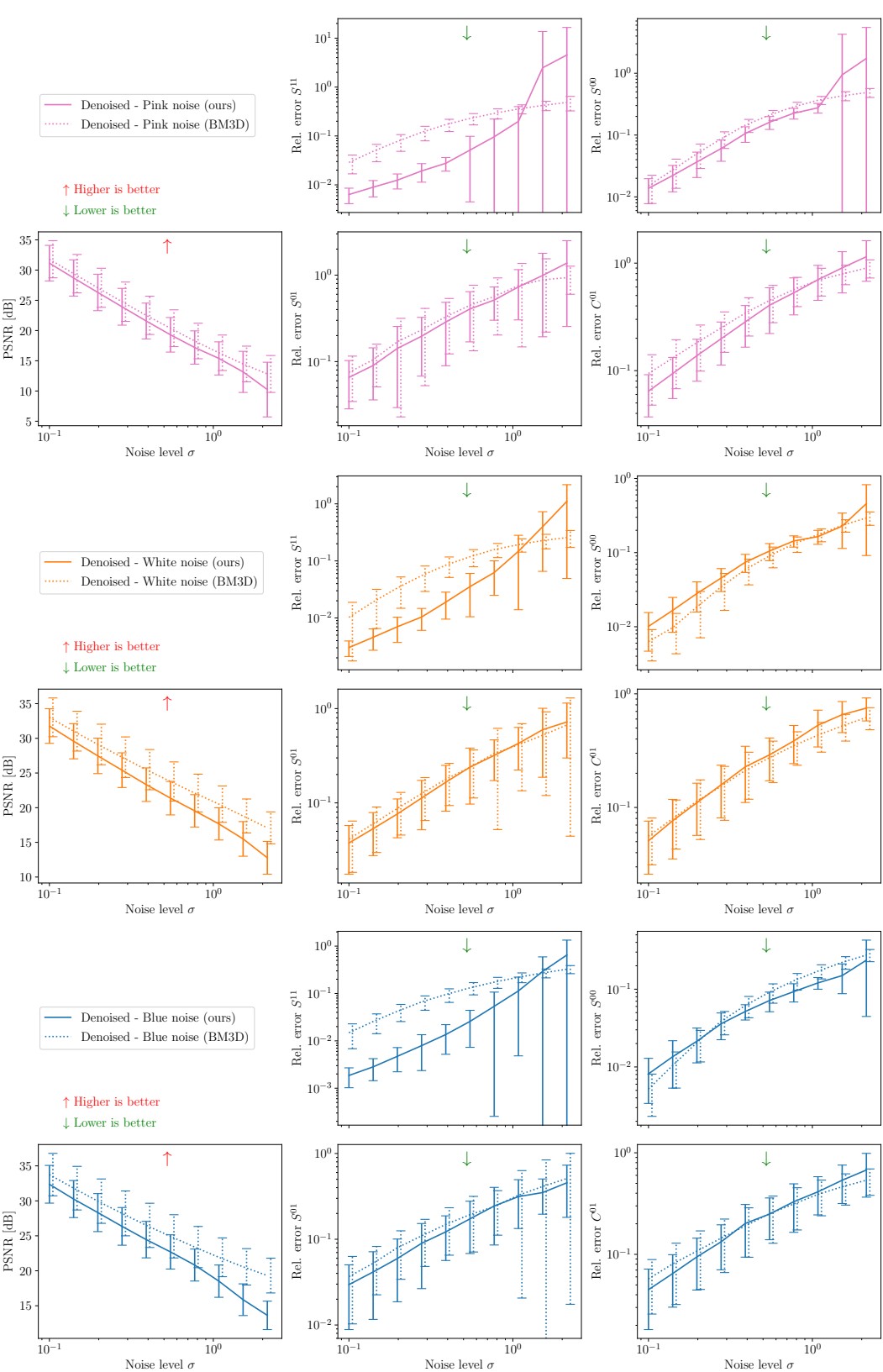

Figure F.7: Same as Fig. 3 for randomly selected ImageNet images. We show the mean and standard deviation of the metrics computed across random batches of test ImageNet images.

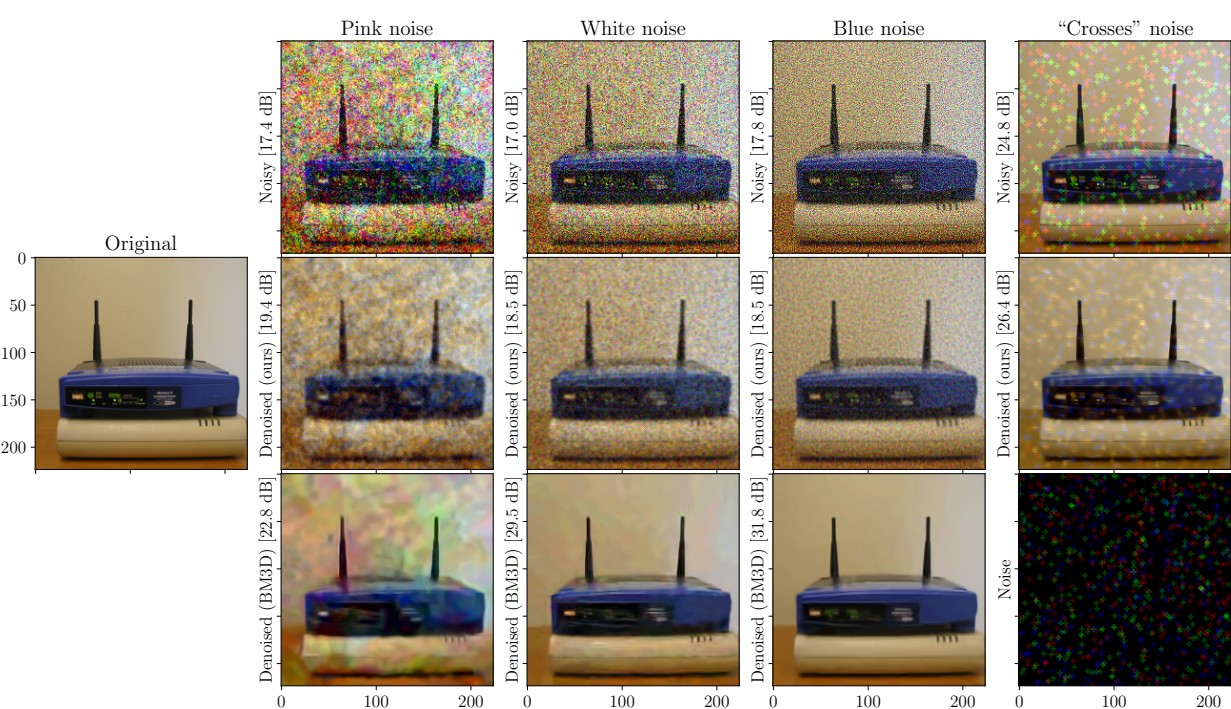

Figure F.8: Same as Fig. 2 for an example of ImageNet image and a ConvNet-based representation (see Sect. 3.3).

