# OpenReview forum: "Statistical Component Separation for Targeted Signal Recovery in Noisy Mixtures"
_TMLR — Accepted by TMLR_

### Review · Reviewer_ZTZn · 2023-11-20

**Summary Of Contributions:**

This paper aims to solve the problem of “statistical component separation” that focuses on recovering a predefined set of statistical descriptors of a target signal from a noisy mixture. Assuming prior knowledge of the noise distribution, the authors analyze a method devised to match the statistical discriptor of the solution. For simple discriptors (such as linear, quadratic, and power spectrum representations), analytic solutions are provided after simple calculations. The authors also empirically studied 1) wavelet-based descriptors and 2) ConvNet-based descriptors, and showcase the performance of their method in recovering (from noisy measurement) of the underlying noise-free discriptors. The authors also extend the method by introducing a diffusive stepwise algorithm, with slightly improved denoising performance.

**Audience:**

No

**Claims And Evidence:**

Yes

**Requested Changes:**

1. In proposition 2.3, is $A$ injective? Isn't $A$ the convolution with bandpass filters, and hence not injective?

**Strengths And Weaknesses:**

**Strength**
1. The paper is well-written and well-organized. It is easy to follow.
2. The paper seems to be technically correct. Candid numerical experiments and comparison are provided to illustrate the performance of the proposed method.

**Weakness**
1. The significance of the paper is questionable. The proposed method has only been studied by the same set of authors in recent years, and I wonder what is the difference/relation between the proposed method and maximum-likelihood?
2. I understand the method is not supposed to be a denoising algorithm, so one does not expect it to beat BM3D in denoising. However, I wonder what "is" the strong suit of the proposed method? In particular, after obtaining the WPH and VGG statistics (with a slightly better relative error), what can I use it for?
3. The proposed diffusive algorithm only slightly improves the performance, but might introduce much more computational cost.

---

> ### Author Response · Authors · 2024-01-05
> **Response to Reviewer ZTZn**
>
> We thank the reviewer for their helpful comments.
>
> Concerning the weaknesses:
> 1. We regret that the reviewer raises doubts on the significance of the paper. While it is true that the proposed method has been introduced in the literature only recently and employed by intersecting sets of authors, we do not believe this should discredit our work. Moreover, while the past literature focused on astrophysical applications of the method, our paper takes a very new direction in the sense that it addresses a first formal and systematic exploration of the method in itself. We believe that this novelty meets the acceptance criteria of TMLR publications.
>
> Concerning the difference/relation of our method with maximum likelihood estimation (MLE), we have now complemented the paper with an additional appendix (App. B in the revised manuscript) that makes a first attempt to relate our method to conventional MLE methods in the contexts of both features estimation and denoising. For features estimation, MLE requires an additional model relating the features $\phi(x_0)$ to $x_0$. We discuss the case of a macrocanonical maximum entropy model and explain that the likelihood remains generally intractable, which prevents further connections. For denoising, our method shares some similarities with MLE (or, more precisely, the estimation of a maximum a posteriori) in the sense that it includes both a constraint of proximity to the observation $y$ and a regularization constraint materialized by the loss of Eq. (1).
>
> 2. The strong suit of the method is contexts where the noise distribution is non-Gaussian (where standard denoisers are not applicable) and for which we only have access to a given set of samples/examples. Being able to estimate arbitrary features of the target signal based on the observation of a noisy mixture can have several applications. In astronomy, the goal can simply be descriptive, i.e. to extract summary descriptive statistics of the target signal, or to build statistical models. WPH descriptors have been used in the past to define generative models of stationary processes (maximum entropy models) in various contexts (see e.g. Zhang & Mallat 2021, Allys et al. 2020, Brochard, Zhang & Mallat 2022).
>
> 3. Indeed, from a numerical perspective, the interest of the diffusive algorithm appears to be limited. However, we argue in the paper that it brings another valuable theoretical perspective on the method.
>
> Concerning the comment of the “Requested Changes” section, we confirm that the matrix $\Psi_i$ as defined in Sect. 2.3. is injective. As explained in the text, we restrict the linear operator representing the bandpass filtering operation to the subspace defined by Fourier eigenmodes spanning the passband of the filter. The resulting operator is bijective by construction.

---

> > ### Comment · Reviewer_ZTZn · 2024-01-15
> >
> > I thank the authors for the detailed response. The added appendix B indeed helps to clarify the relation of the proposed method to MLE. While I have concern on the numerical aspect  on the diffusive algorithm, most of my questions have been resolved.

---

### Review · Reviewer_ShsH · 2023-12-01

**Summary Of Contributions:**

This work analyzes a recently proposed approach to extract signals from noisy mixtures. The signal to be extracted is specified in terms of a set of desired statistics (computed by representation transform $\phi$), presented to the algorithm via a (possibly small) dataset of template signals. This paper analyzes the cost function of this problem for linear $\phi$ and quadratic $\phi$, leading up to analysis of a power-spectrum based summary statistic that has been shown to be effective in practice. The authors provide a statistical explanation of this problem's solution's relationship with the desired signal component (the ground truth used to generate the mixture signal under the assumed model).

Numerical experiments quantitatively compare the algorithm in a denoising setting, and smaller experiments are shown for a "structured noise" (aka component separation with randomly placed crosses) setting.

Finally, a diffusive extension of the algorithm is derived and also compared with BM3D in a white noise denoising setting.

**Audience:**

Yes

**Claims And Evidence:**

Yes

**Requested Changes:**

Critical to securing recommendation:
1. A recurring theme is that this work does an insufficient job presenting its position in the context of optimization, signal separation, & denoising at large. This is illustrated by your conclusion following the statement of Proposition 2.1, that a nonlinear $\phi$ must be used to perform denoising, thereby discarding (literally) hundreds of years of denoising theory, not to mention highly successful, modern signal separation methods that use linear $\phi$[1-3]. So I find this to be a fallacious (and a bit lazy?) justification for your choice to use nonlinear $\phi$. The authors/ the paper/ the readers would be much better served if the authors owned up to the decision from a holistic perspective. For a journal paper, isn't it reasonable to ask: why do you choose to use the regularization function $R(x,y)=0$ in Cost Function (1)? How is the proposed approach to incorporating prior information better than the ``Bayesian'' way (via a choice of $R(x,y)\neq0$)?

I understand that Prop 2.1-2 are in the paper to lead up to Prop 2.3 (which is nicely done!), and I do not expect you to seriously numerically compare these. However they are brushing up against a whole world of techniques that are just barely referenced. In other words, no one would ever just "shrink the signal to zero" if the noise is just over said threshold. There are iterative methods and many continuous distributions of thresholding operators between soft thresholding and identity (see [4] and Fig 1 in [5]).

Proposition 2.1: Why not allow $E[\epsilon]\neq0$ show that you get a biased result? For component separation (as opposed to Gaussian denoising), of course $E[\epsilon]$ would be nonzero.

2. In my opinion, you should either (1) change all mention of "component separation" and "mixtures" to focus on what this is: denoising. You have a nice denoising analysis for different colors of Gaussian noise but only one single realization of the crosses noise. At the very least there should be some analysis on a series of corruption levels as in Figs 3-5. Or, (2), provide additive mixture examples where $\epsilon$ is not just Gaussian noise. I think this paper would be so much stronger if it stood against other component separation approaches (MCA is the main one that comes to mind [1-3]). How, and in what sense, must $\epsilon$ be distinct from $x_0$ for successful extraction? Your statistical analysis may be able to address this critical question that is also outstanding in MCA (see "spark" in [3]).

Other suggestions:
* Isn't $\mathbb{K}=\mathbb{C}$ ? Why don't you just use $\mathbb{C}$?
* A true novelty of statistical component separation compared to so much convex optimization literature is in the way a dataset (e.g. a way to sample \epsilon) is used to shape the optimization landscape... and a novelty compared to other ML approaches is the possibility of using relatively small datasets. I think it will be more clear to the majority of readers (who just glance at the first page or two) if you state the empirical cost function ($\hat{L}$ in Alg 1) near the beginning with (1)
* The work performed in Lemma D is perhaps worth emphasizing \textit{more}, e.g. providing a framework that joins the approaches.
* Instead of picking a "random" image from imagenet (which I find hard to believe anyway...), why not perform the experiment on N random draws of imagenet and thus providing an interesting statistically significant experiment?
* There are a number of places where the quotation marks do not match on either side of the word. Typically in LaTeX it looks good to have `` on the left and '' on the right (two apostrophes, not one right-quotation mark ")

[1] Goehle, Geoff, et al. "Approximate extraction of late-time returns via morphological component analysis." The Journal of the Acoustical Society of America 153.5 (2023): 2838-2838.

[2] Parekh, Ankit, et al. "Multichannel sleep spindle detection using sparse low-rank optimization." Journal of neuroscience methods 288 (2017): 1-16.

[3] Starck, J-L., Michael Elad, and David L. Donoho. "Image decomposition via the combination of sparse representations and a variational approach." IEEE transactions on image processing 14.10 (2005): 1570-1582.

[4] Selesnick, Ivan. "Penalty and shrinkage functions for sparse signal processing." Connexions 11.22 (2012): 23.

[5] A. H. Al-Shabili, Y. Feng and I. Selesnick, "Sharpening Sparse Regularizers via Smoothing," in IEEE Open Journal of Signal Processing, vol. 2, pp. 396-409, 2021, doi: 10.1109/OJSP.2021.3104497.

**Strengths And Weaknesses:**

Strengths:
* The analysis aspects of the paper are clear, and a contribution to the community.
* The overall language of the paper is very clear.
* The experiments are fairly designed; both strengths and weaknesses of the proposed algorithm are portrayed.

Weaknesses:
1. Some language in this work seems to forget its position in the broader context of signal separation / denoising / optimization at large. Details are in 'requested changes' section.
2. One specific example (with only one realization) of structured noise is provided (randomly placed crosses), i.e., the rightmost column of Fig 2. Everything else in the paper is _denoising_, *not* _signal component separation_. The paper presents itself as a "component separation" algorithm that works with "mixtures", but that's totally misleading when the only undesired component is Gaussian noise.

---

> ### Author Response · Authors · 2024-01-05
> **Response to Reviewer ShsH**
>
> We thank the reviewer for their detailed and helpful review. We give responses below for each of the comments raised in the “Requested Changes” section.
>
> **Critical to securing recommendation:**
>
> 1. We understand the impression of the reviewer that our work does an insufficient job at situating the proposed method in the landscape of optimization, signal separation, and denoising. However, we first want to clarify that this paper focuses on the analysis of a very specific method, which takes an orthogonal direction to the literature mentioned by the reviewer (in its goal and also partially in its means). The goal of the method is not to estimate $x_0$ (as in a classical denoising context), but to estimate $\phi(x_0)$. which in particular does not necessarily require proximity of the solution $\hat{x}_0$ to $y$. Of course, our method can still be interpreted in a denoising context, but this was not the primary goal of the paper. Nevertheless, we have added in App. B an interpretation of our method within this context and from the perspective of the estimation of a maximum a posteriori image. From that picture, the loss $\mathcal{L}$ can be seen as a regularization term, where the term enforcing proximity to $y$ in signal space has been neglected. It remains however difficult at this stage to argue whether this choice of regularizer is more relevant than another one for a denoising purpose (especially as the $\phi$ function remains arbitrary).
>
> While Prop. 2.1's outcome may appear surprising in the context of denoising, it stands true as stated. To clarify, this result is not a superficial justification for the choice of a nonlinear $\phi$, but a demonstration that the optimum of the loss defined in (1) is trivial for linear $\phi$. We have rephrased the paragraph to improve clarity. Also, as suggested by the reviewer, we now present a slightly more general form of this proposition allowing for non-zero mean of the noise.
>
> Concerning Prop. 2.2 and 2.3, we thank the reviewer for the very interesting references which put in perspective these results (these are now mentioned in Sect. 2.2). We also want to clarify that the propositions are again very specific to the loss function (1), and we are not recommending the shrinking of the signal when the noise goes beyond a given threshold, but just saying that this is what will happen with our method in that context.
>
> 2. We have conducted new experiments on the “crosses” noise as suggested. We analyzed the performance of our method in the case of the “crosses” noise as a function of the density of crosses. This analysis demonstrates a clear mitigation of the noise in all regimes.
>
> A comparison of our method to other component separation approaches such as MCA would be very interesting, but we prefer to leave such a study to further work.
>
> Concerning the very interesting question of how distinct must $\epsilon_0$ be from $x_0$ for successful extraction, it remains a very hard question that cannot be addressed generally as it directly depends on the choice of $\phi$. As an illustration, Prop. 2.3 shows that we can get perfect reconstruction provided that $x_0$ and $\epsilon_0$ live on disjoint frequency bands for $\phi$ a power spectrum representation with consistent frequency bands. However, addressing more general cases than these ones remains very difficult.
>
> **Other suggestions**
>
> 1. Depending on the contexts, $\mathbb{K}$ could be either $\mathbb{R}$ or $\mathbb{C}$, but to simplify we now only mention $\mathbb{K}=\mathbb{C}$ in the “Notations” section.
>
> 2. We have added the expression of the empirical loss function straight from the introduction to clarify this point.
>
> 3. We have put a little more emphasis on the results of App. D (which became App. E in the revised version) in the main text.
>
> 4. For the analyses of this paper, it was easier for us to reason on a limited set of images $x_0$, providing us greater flexibility to explore variability in relation to noise. We confirm that the ImageNet image was selected randomly (note that our ImageNet results don’t particularly put our method in the best light). Nevertheless, as suggested by the reviewer, we have conducted a new set of experiments involving random batches of ImageNet data ($N=50$), and added the results to the paper (cf updated Fig. 5 and new Fig. F.7). In most settings, our method appears to be better suited than BM3D to recover the features $\phi(x_0)$ for both the WPH and VGG representations.
>
> 5. Thank you for spotting this, we have corrected the quotation marks.
>
> 6. Could the reviewer clarify what they meant by “[tone mapper]”?

---

> ### Comment · Reviewer_ShsH · 2024-01-07
> **Satisfied with changes**
>
> Thank you for your responses, explanations, and expansions in the paper. I appreciate the extra effort and time. I am happy with the results and recommend the paper for publication. I hope to see this method expanded further and compared with other approaches for separating complex data in the future.
>
> PS: never mind #6, that was a reminder for myself to look into something that your method reminded me of, I just forgot to erase it. Apologies.

---

### Review · Reviewer_bxtv · 2023-12-10

**Summary Of Contributions:**

This paper analyzes and extends the "statistical component separation" algorithm developed by Regaldo-Saint Blancard et al. and others. The algorithm assumes one has a single noisy observation, y=x+\epsilon, of a signal contaminated with noise and a method for sampling from the distribution of the noise. It then seeks a solution \hat{x} that minimizes that minimizes the expected l2 difference between \phi(x+noise) and \phi(y), where \phi is user-defined function and y is the noisy observation. Previous works have empirically demonstrated the resulting \hat{x} and \phi(\hat{x}) are effective estimates of x and \phi(x).

This paper...
-Provides several examples of operators \phi() where the above optimization problem has interesting analytic solutions.
-Provides empirical evidence that with \phi() defined as the covariances of wavelet coefficients of the images, the proposed method effectively denoises images (particular atmospheric images of dust) and can handle complex noise distributions (crosses randomly added to locations on the image).
-It demonstrates that (somewhat counter-intuitively) defining \phi() as the VGG feature mapping and matching the signal there results in poor overall denoising performance. I think this is an interesting negative result.
-It develops (and partially analyzes) a diffusion-inspired denoising method which gradually denoises the signal in stages. This approach marginally improves performance over the single-shot approach.

**Audience:**

Yes

**Broader Impact Concerns:**

No concerns.

**Claims And Evidence:**

Yes

**Requested Changes:**

The optimizer was only run for 30 iterations. Please comment on the effect the number of iterations has on the performance of the algorithm. Is early stopping (à la deep image prior) responsible for some of the denoising performance?

**Strengths And Weaknesses:**

## Strengths
Overall, I think this paper is a good fit for TMLR. It analyzes an interesting recently-developed algorithm that could have a significant impact in astronomy and other scientific domains where data is hard to come by. While the performance of  extensions proposed in the paper is somewhat disappointing, I think this represents an interesting negative result that will inform the ML community.

The paper is relatively easy to follow.

## Weaknesses
The paper might benefit from a high-level comparison with maximum likelihood estimation and other conventional estimation methods.
The performance of the proposed methods is underwhelming.

---

> ### Author Response · Authors · 2024-01-05
> **Response to Reviewer bxtv**
>
> We thank the reviewer for their positive and helpful comments.
>
> Concerning the weaknesses:
> 1. We have now complemented the paper with an additional appendix (App. B) that makes a first attempt to relate our method to conventional maximum likelihood estimation (MLE) methods in the contexts of both features estimation and denoising. For features estimation, MLE requires an additional model relating the features $\phi(x_0)$ to $x_0$. We discuss the case of a macrocanonical maximum entropy model and explain that the likelihood remains generally intractable, which prevents further connections. For denoising, our method shares some similarities with MLE (or, more precisely, the estimation of a maximum a posteriori) in the sense that it includes both a constraint of proximity to the observation $y$ and a regularization constraint materialized by the loss of Eq. (1).
> 2. Indeed, the numerical analyses of this paper exhibit mixed results in the case of Gaussian noises for both denoising and features estimation in comparison to BM3D. However, we stress that there are clear situations where our method systematically outperforms BM3D, e.g. the estimation of power spectrum-like features (cf the $S^{11}$ coefficients of the WPH representation). Correctly recovering the power spectrum of the target signal is often a priority for scientific applications, and our analysis raises awareness on the fact that regular denoisers might not be suited for this purpose. But more importantly, our method applies beyond the Gaussian noise setting and for arbitrary representations, where regular denoisers are not applicable. Since an important part of this paper is dedicated to benchmarking our method in a denoising context, we mostly restricted our analysis to Gaussian noises where we could compare our method to standard denoisers.
>
> Concerning the comment in the “Requested Changes” section, we want to clarify that for the experiments of this paper, increasing the number of iterations of the optimizer does not impact the results. Choosing $T=30$ optimization steps was sufficient to reach approximate convergence in all experiments. In comparison, the choice of the noise batch size $Q$ has been much more decisive, since for a too small value of $Q$, the performance of our method could deteriorate. We have clarified this point in the text.

---

### Author Response · Authors · 2024-01-05
**General Response to the Reviewers**

We would like to thank all reviewers for their helpful comments, which enabled us to clarify several aspects of the paper. In addition to the individual responses, we summarize here the main extensions and revisions to our work that resulted from the reviews.

First, we made new experiments. As suggested by Reviewer ShsH, we ran two additional sets of experiments:
1. We analyzed the performance of our method in the case of the “crosses” noise as a function of the density of crosses (cf Figs. 4, F.3, F.6 of the revised manuscript). This analysis demonstrates a clear mitigation of the noise in all regimes.
2. We have rerun our analyses on a set of randomly selected images from the ImageNet dataset (batches of $N=50$ images). The corresponding quantitative results show average metrics across these randomly selected images (cf Figs. 5 and F.7 of the revised manuscript). In most settings, our method appears to be better suited than BM3D to recover the features $\phi(x_0)$ for both the WPH and VGG representations.

We also fixed a minor typo in our plotting scripts, leading to updates of some of our figures (with no impact on the  discussion).

Second, we made several edits to the text following the recommendations and concerns of the reviewers. These are marked in red in the revised submission. In particular, since a recurring comment of the reviewers was the insufficient comparison of our method with standard denoising or estimation techniques, we added a new appendix to the paper (App. B) that discusses the relation between our method and maximum likelihood estimation methods (interpreted in a broad sense). This appendix gives first elements to better situate our method in the landscape of denoising techniques. However, we want to emphasize that denoising is not the primary goal of our method, which rather focuses on the estimation of the features $\phi(x_0)$. We also provide in App. B a first attempt to connect our method to MLE when the goal is to estimate $\phi(x_0)$.

---

### Decision · Action_Editor_3Z6H · 2024-01-29

**Recommendation:** Accept with minor revision

**Comment:**

Overall, all three reviewers agreed that the paper had interesting contributions, and hence were positive about publishing it. After looking at the paper myself, I agree that this is an interesting idea.

For minor revisions, please do a final check for typos, and fix a few capitalization issues in the bibliography (e.g., capitalize "herschel" since it is a proper noun).

**Audience:**

The audience is broad enough for TMLR, as signal recovery is a very wide topic, and even this specific scenario is reasonably broad. The technique has so far been used mostly in astrophysics. By publishing in a more general-purpose venue like TMLR the technique may see wider applicability (or statisticians may be able to show how this technique relates to other existing techniques -- either way, it's valuable).

**Claims And Evidence:**

The authors consider a fairly generic signal recovery problem, with the twist that they only want to recover a function of the original signal. The core idea was developed in a few papers since 2021 (mostly written by an overlapping subset of authors), and this paper extends those ideas in a few interesting directions.  The paper has some theory and numerical computations; reviewers were positive about the general setup and the technical correctness of the theory and computations.

---

> ### Author Response · Authors · 2024-02-28
>
> Dear Editor and Reviewers, we sincerely appreciate the time and effort you dedicated to reviewing our paper. We are honored by the acceptance decision and grateful for the insightful feedback and constructive suggestions provided throughout the review process. We will submit the camera ready version of the manuscript shortly, incorporating the minor revision suggested by the editor.